# AbFlowNet: Optimizing Antibody-Antigen Binding Energy via Diffusion-GFlowNet Fusion

## Abstract

Complementarity Determining Regions (CDRs) are critical segments of an antibody that facilitate binding to specific antigens. Current computational methods for CDR design utilize reconstruction losses and do not jointly optimize binding energy, a crucial metric for antibody efficacy. Rather, binding energy optimization is done through computationally expensive Online Reinforcement Learning (RL) pipelines rely heavily on unreliable binding energy estimators. In this paper, we propose AbFlowNet, a novel generative framework that integrates GFlowNet with Diffusion models. By framing each diffusion step as a state in the GFlowNet framework, AbFlowNet jointly optimizes standard diffusion losses and binding energy by directly incorporating energy signals into the training process, thereby unifying diffusion and reward optimization in a single procedure. Experimental results show that AbFlowNet outperforms the base diffusion model by $3.06\%$ in amino acid recovery, $20.40\%$ in geometric reconstruction (RMSD), and $3.60\%$ in binding energy improvement ratio. ABFlowNet also decreases Top-1 total energy and binding energy errors by $24.8\%$ and $38.1\%$ without pseudo-labeling the test dataset or using computationally expensive online RL regimes. [1]

## 1 Introduction

Antibodies are essential molecules of the adaptive immune system, with their complementarity-determining regions (CDRs) serving as the primary determinants of antigen recognition and binding specificity. Compared to the rest of the antibody, CDRs exhibit remarkable variability, enabling the immune system to recognize diverse antigens (Polonelli et al., 2008). CDRs are crucial for therapeutic antibody development, particularly in *humanization* where CDRs from non-human antibodies are transferred onto human antibodies to help it target new antigens, using techniques like CDR grafting (Jones et al., 1986) and shuffling (Jirholt et al., 1998). Daclizumab, the first FDA-approved humanized drug, was developed in 1997 by humanizing a mouse antibody to treat multiple sclerosis (Tsurushita et al., 2005). Since then, thousands of other drugs have been developed using CDR-based antibody modifications (Lu et al., 2020).

GFlowNet Trajectory Balance Objective: $Z_\theta \prod_{t=1}^{n} P_F(s_t | s_{t-1}) = R(s_n) \prod_{t=1}^{n} P_B(s_{t-1} | s_t)$

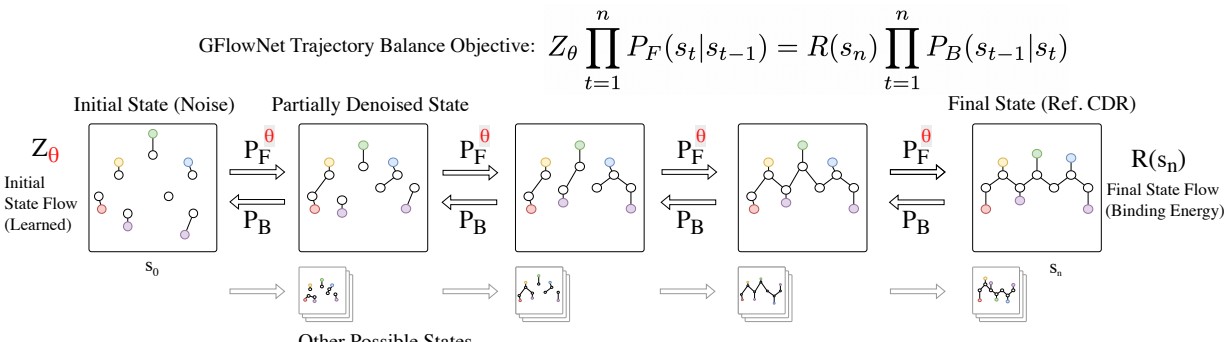

Figure 1: AbFlowNet reframes the diffusion process as a GFlowNet where each partially denoised CDR is a *state* and the transition probabilities are *flows through edges*. The initial state's flow is learned and the final state's flow is the binding energy of the reference CDR. To train, we simply enforce forward and backward flow parity, in addition to the diffusion losses.

[1]The code and model weights are available at `anonymous.4open.science/r/abflownet-06E8`.

While transferring known animal CDRs has proven effective, there has been immense research interest into designing *de novo* CDRs to target novel antigens (Tang et al., 2024) and neoantigens (Zhang et al., 2021). The computational (*in silico*) design of CDRs presents a significant challenge due to the vast search space - a CDR sequence with $L$ amino acids has $20^L$ possible combinations, not accounting for structural variations. Traditional Monte Carlo search-based approaches use biophysical energy functions (Adolf-Bryfogle et al., 2018a; Lapidoth et al., 2015; Adolf-Bryfogle et al., 2018b) to guide the search process but are computationally intensive and often get trapped in local optima (Luo et al., 2022; Jin et al., 2022). Deep Learning (DL) approaches using either Graph Neural Networks (Gao et al., 2023; Jin et al., 2022; Kong et al., 2023; 2022) or Diffusion models (Luo et al., 2022; Peng et al., 2023; Zhu et al., 2024) can learn the distribution of existing CDRs and sample new ones. However, unlike search-based approaches, DL methods do not explicitly optimize biophysical energy functions which has led to new research on online Reinforcement Learning (RL) (Sutton et al., 1998) post-training using these functions as rewards (Zhou et al., 2024; Ren et al., 2025; Wen et al., 2024). However these online RL regimes are extremely computationally expensive and highly dependent on the programs used to estimate said energy functions which are not always reliable (Vreven et al., 2012). Some approaches (Zhou et al., 2024; Wen et al., 2024) use the test dataset itself during RL which raises concerns about dataset bias (Deng et al., 2023). Furthermore, Zhou et al. (2024) has shown that RL improves binding energy but reduces two key structural metrics: Amino Acid Recovery and Root Mean Square Deviation with respect to the reference CDR.

Given the limitations of RL, GFlowNets have emerged as a promising alternative for optimizing Image Diffusion models (Venkatraman et al., 2024; Zhang et al., 2024a;b). In this work, we introduce AbFlowNet to address these concerns in biophysical energy optimization. As shown in Figure 1, we reframe the denoising diffusion process (Ho et al., 2020; Leach et al., 2022) in a GFlowNet (Bengio et al., 2021) framework where each partially denoised structure is a GFlowNet *state* and the forward and backwards transition probabilities are *flows through edges*. The *flow of a state* is the sum of all flow's of all trajectories through that state. The final fully-denoised state's flow is the binding energy *reward*. We use the Trajectory Balance objective (Bengio et al., 2021) to enforce that the forward and backward flow for a trajectory (a full denoising sequence starting from random noise and ending at a CDR structure) must be equal. As a result, the diffusion model implicitly learns better state transitions that lead to higher rewards.

Practically, AbFlowNet can be implemented by adding a single learned parameter and adding a TB loss term to the original loss terms. AbFlowNet shows convincing improvements over the base diffusion model, DiffAb (Luo et al., 2022), for the same number of gradient updates. Concretely, when averaged over all six CDR regions, AbFlowNet improves amino acid recovery (AAR) by $3.06\%$, root mean square deviation (RMSD) by $20.40\%$ and samples $3.60\%$ more CDRs that have better binding energy than the reference CDR. AbFlowNet also improves over DiffAb in Top-1 total energy and binding energy by $24.8\%$ and $38.1\%$. Unlike online RL approaches such as AbDPO (Zhou et al., 2024), AbFlowNet is orders-of-magnitude less expensive, does not need repeated use of unreliable energy estimators and does not rely on pseudo-labeling the test set. Our key contributions are:

1. We present AbFlowNet, the first application of the GFlowNet framework for direct binding energy optimization in *de novo* diffusion-based CDR design. AbFlowNet improves over the base diffusion model in all metrics.

2. AbFlowNet is competitive with RL-based methods (Zhou et al., 2024) without using the test set complexes to generate synthetic CDR data for training, thereby mitigating data leakage concerns.

3. Unlike existing RL-based approaches (Zhou et al., 2024; Wen et al., 2024) which reduce AAR and RMSD, AbFlowNet improves AAR by $+3.06\%$ and RMSD by $+20.40\%$.

## 2 RELATED WORKS

**Computational CDR Design**  Classical approaches to CDR design, such as RAbD Adolf-Bryfogle et al. (2018a) and AbDesign Lapidoth et al. (2015), rely on Monte Carlo algorithms that sample and optimize antibody structures based on biophysical energy functions. These methods, while effective in certain contexts, are computationally expensive and often get trapped in local optima due to the rugged energy landscape (Luo et al., 2022; Kong et al., 2023). In recent years, deep learning methods have emerged as promising alternatives. Notable Graph Neural Network-based models include HERN (Jin et al., 2022), MEAN Kong et al. (2022), AbGNN (Gao et al., 2023) and dyMEAN (Kong et al., 2023). This methods have shown high AAR and RMSD but are limited in generation diversity due to their GNN structure. Diffusion-based models can generate multiple CDR given a complex which can be later ranked heuristically. Notable Diffusion-based models include AbDiffuser (Martinkus et al., 2024), DiffAb (Luo et al., 2022), AbDesign (Peng et al., 2023), and AbX (Zhu et al., 2024). AbX is the current state-of-the-art CDR design model and uses a large Protein Language Model (Lin et al., 2022) to enforce evolutionary plausibility of the generated CDRs.

**Binding Energy Optimization For CDR Design**  A key metric for CDR effectiveness is binding affinity. One commonly used energy metric is binding energy $\Delta G$. Since binding energy is a singular value for the entire complex,

it is a sparse training signal which is often optimized via Reinforcement Learning (Sutton et al., 1998). AbDPO (Zhou et al., 2024) post-trains a base DiffAb model by repeatedly sampling new CDRs, ranking them based on binding affinity, determined with `Rossetta InterfaceAnalyzer` (Chaudhury et al., 2010) and using DPO (Rafailov et al., 2024). However, this RL training phase significantly lowers AAR and RMSD compared to the base method. AlignAb (Wen et al., 2024) points out that there are multiple valid energy-based rewards and finetune separate models for each reward using DPO. AbNovo (Ren et al., 2025) follows the approach of AbDPO with AbX instead of DiffAb as the base model and used Noise Contrastive Alignment (NCA) (Chen et al., 2024) as the RL objective instead of DPO.

One notable weakness of all online RL methods is the need to compute binding energy for newly designed CDRs. *In silico* methods such as `Rosetta` (Chaudhury et al., 2010; Adolf-Bryfogle et al., 2018a) or `OpenMM Yank` (Eastman et al., 2017; Rizzi et al., 2020) have only moderate corelation with the real binding energy(Vreven et al., 2012). Furthermore, the generated CDRs are not guaranteed to be geometrically plausible which might reduce the reliability of energy estimators further. In contrast, AbFlowNet does not require computing the energy of newly generated CDRs and can, in principle, be trained solely on *in vitro* affinity data of CDRs in the training set.

## 3 BACKGROUND

### 3.1 ANTIBODY-ANTIGEN COMPLEX

As shown in Figure 2, antibodies are composed of two heavy chains and two light chains. Each chain consists of a variable region and a constant region. The variable regions of both the heavy chain ($V_H$) and the light chain ($V_L$) contain three complementarity determining regions (CDRs): CDR1, CDR2, and CDR3, making a total of six CDRs per antibody. These regions are highly diverse due to genetic recombination and somatic hypermutation, allowing antibodies to recognize a vast array of antigens. The CDRs form a binding site that is complemen-

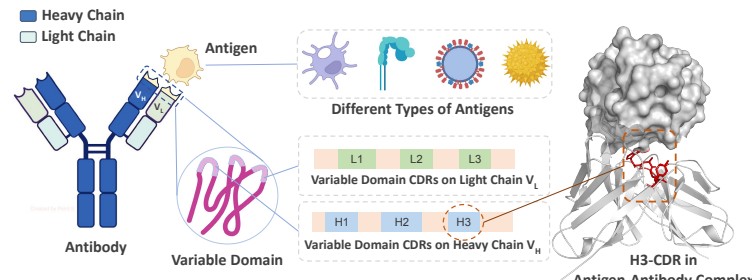

Figure 2: Antibody-antigen complex.

tary in shape and chemical properties to the antigen's binding site (epitope). In Figure 2, the third CDR in the heavy chain (CDR-H3) is highlighted due to its critical in determining the binding affinity to the antigen. The sequence and structure of CDRs vary widely among antibodies, enabling the immune system to recognize and respond to a wide range of antigens. CDRs interact with the antigen through non-covalent bonds (e.g., hydrogen bonds, electrostatic interactions, van der Waals forces) (Polonelli et al., 2008). In our work, we aim to design the sequences and structures of the CDR regions (often referred to as the framework regions), conditioned on the non-CDR regions of the antibody and on the target antigen.

### 3.2 GFLOWNET

GFlowNets are generative models that learn to sample from a desired distribution by modeling flows on a directed acyclic graph (DAG) (Bengio et al., 2021). Given a DAG $G = (\mathcal{S}, \mathcal{A})$ with state space $\mathcal{S}$ and action space $\mathcal{A}$, and a positive reward function $R : \mathcal{X} \to \mathbb{R}_{\geq 0}$ defined on terminal states $\mathcal{X}$, a GFlowNet learns a policy that generates trajectories terminating at states with probability proportional to their rewards. Formally, a GFlowNet defines a flow $F$ on trajectories $\tau = (s_0 \to s_1 \to ... \to s_n)$ from the initial state $s_0$ to terminal states. The ***state flow*** of state $s$ is defined as $F(s) = \sum_{\tau=(...\to s...)} F(\tau)$ and the ***edge flow*** between states $s$ and $s'$ is defined as $F(s \to s') = \sum_{\tau=(...\to s \to s'...)} F(\tau)$. The following *flow matching* constraint (incoming flow = outgoing flow) is satisfied for all non-boundary states $F(s) = \sum_{(s'' \to s) \in \mathcal{A}} F(s'' \to s) = \sum_{(s \to s') \in \mathcal{A}} F(s \to s')$. For terminal states $s_n$, the flow is the non-negative reward: $F(s_n) = R(s_n)$.

Flows also induce forward and backward transition policies: $P_F(s'|s) = \frac{F(s \to s')}{F(s)}$ and $P_B(s|s') = \frac{F(s \to s')}{F(s')}$. A GFlowNet aims to learn policies such that the terminal state flows match their rewards.

**Trajectory Balance** Trajectory Balance (TB) (Malkin et al., 2022) provides an elegant training objective for GFlowNets that enforces consistency between forward generation and backward reconstruction across entire trajec-

tories. For any complete trajectory $\tau = (s_0 \rightarrow s_1 \rightarrow \ldots \rightarrow s_n)$ terminating at state $x$, TB enforces the constraint:

$$F(s_0) \prod_{t=1}^{n} P_F(s_t|s_{t-1}) = F(s_n) \prod_{t=1}^{n} P_B(s_{t-1}|s_t) \tag{1}$$

Here, the flow of the terminal state $s_n$ is $F(s_n) = R(s_n)$. The flow of the initial state $F(s_0)$ is the sum of the flow over all trajectories which is not tractable. Therefore, the authors of TB propose approximating $F(s_0)$ with $Z_\theta$ where $\theta$ is a neural network. This yields the final constraint:

$$Z_\theta \prod_{t=1}^{n} P_F(s_t|s_{t-1}) = R(s_n) \prod_{t=1}^{n} P_B(s_{t-1}|s_t) \tag{2}$$

## 4 METHODOLOGY

In Section 4.1, we discuss the training objectives of the base diffusion model, and in Section 4.2, we describe our reframing of the diffusion process as a GFlowNet. Equation 18 presents our final training objective, which jointly optimizes the diffusion losses and the binding energy.

### 4.1 DENOISING OBJECTIVE

We train a diffusion probabilistic model parameterized by a neural network for CDR design. Following Luo et al. (2022), we condition on the antigen structure and the antibody framework to generate CDR. The model is trained using the standard denoising objective across three protein properties: **amino acid type** $d_i \in \{A, C, D, E, F, G, H, I, K, L, M, N, P, Q, R, S, T, W, Y, V\}$, **3D coordinate** $x_i \in \mathbb{R}^3$, and **3D orientation** $O_i \in SO(3)$ where $SO(3)$ is the Lie group of 3D rotations.

Assume the CDR to be generated has $m$ amino acids with index from $l+1$ to $l+m$. They are denoted as $S^t = \{s_j^t | j = l+1, \ldots, l+m\}$ where $s_j^t = (d_j^t, x_j^t, O_j^t)$. Our goal is to model the distribution of $S^0$ given the structure of the antibody-antigen complex $C = \{(d_i, x_i, O_i) \mid i \in \{1, \ldots, N\} \setminus \{l+1, \ldots, l+m\}\}$.

**Multinomial Diffusion for Amino Acid Types** The forward diffusion process for amino acid types is based on the multinomial diffusion process (Hoogeboom et al., 2021):

$$q(d_j^t|d_j^{t-1}) = \text{Mul}\left((1 - \beta_{\text{type}}^t) \cdot \text{oh}(d_j^{t-1}) + \frac{\beta_{\text{type}}^t}{20} \cdot \mathbf{1}\right) \tag{3}$$

$$p(d_j^{t-1}|S^t, C) = \text{Mul}\left(F_\theta(S^t, C)[j]\right) \tag{4}$$

where $p$ is the forward diffusion process, $q$ is the backward denoising process, Mul() is the Multinomial function and oh() is the one-hot function. $\beta_{\text{type}}^t$ is the probability of resampling another amino acid uniformly over the 20 types and $F_\theta(\cdot)[j]$ is a neural network model that predicts the probability of the amino acid type for the $j$-th amino acid on the CDR. The training objective is to minimize the expected KL divergence between the posterior distribution $q$ and the predicted distribution $p$:

$$L_{\text{type}}^t = \mathbb{E}\left[\frac{1}{m} \sum_j D_{\text{KL}}\left(q(d_j^{t-1}|d_j^t, d_j^0)\|p(d_j^{t-1}|S^t, C)\right)\right] \tag{5}$$

**Diffusion for 3D Coordinates** The forward and backward diffusion processes for the coordinate $x_j$ are defined as:

$$q(x_j^t|x_j^{t-1}) = \mathcal{N}\left(x_j^t \middle| \sqrt{1 - \beta_{\text{pos}}^t} \cdot x_j^{t-1}, \beta_{\text{pos}}^t I\right) \tag{6}$$

$$p(x_j^{t-1}|S^t, C) = \mathcal{N}\left(x_j^{t-1} \middle| \mu_p(S^t, C), \beta_{\text{pos}}^t I\right) \tag{7}$$

$$\mu_p(S^t, C) = \frac{1}{\sqrt{1 - \beta_{\text{pos}}^t}}\left(x_j^t - \frac{\beta_{\text{pos}}^t}{\sqrt{1 - \tilde{\alpha}_{\text{pos}}^t}} G_\theta(S^t, C)[j]\right) \tag{8}$$

where $\beta_{\text{pos}}^t$ controls the rate of diffusion in $q$. The denoising diffusion process $p$ uses the reparameterization trick (Ho et al., 2020) and $G_\theta(\cdot)[j]$ is a neural network that predicts the standard Gaussian noise $\epsilon_j \sim \mathcal{N}(0, I)$ added instead of predicting $x_j^{t-1}$ directly where $\tilde{\alpha}_{\text{pos}}^t = \prod_{t=1}^T (1 - \beta_{\text{pos}}^t)$.

The training objective is to minimize the expected MSE between $G$ and $\epsilon$:

$$L_{\text{pos}}^t = \mathbb{E}\left[ \frac{1}{m} \sum_j \|\epsilon_j - G(S^t, C)[j]\|^2 \right] \tag{9}$$

**SO(3) Denoising for Amino Acid Orientations**  Following Leach et al. (2022); Luo et al. (2022), the denoising process for orientation directly attempts to predict the final orientation $O_j^0$ from $O_j^t$. The transitions are defined as:

$$q(O_j^t|O_j^0) = \mathcal{IG}_{SO(3)}\Big( O_j^t \big| \lambda(\sqrt{\bar{\alpha}_{ori}^t}, O_j^0),\ 1 - \bar{\alpha}_{ori}^t \Big) \tag{10}$$

$$p(O_j^{t-1}|S^t, C) = \mathcal{IG}_{\text{SO(3)}}\left( O_j^{t-1} \Big| H_\theta(S^t, C)[j], \beta_{\text{ori}}^t \right) \tag{11}$$

where $\bar{\alpha}_{ori}^t = \prod_{\tau=1}^t (1 - \beta_{ori}^\tau)$ with $\beta_{ori}^t$ being the variance increased with step $t$, $\mathcal{IG}_{\text{SO(3)}}$ denotes the isotropic Gaussian distribution on $SO(3)$ (Leach et al., 2022) and $\lambda(\gamma, x) = \exp\big( \gamma \log(x) \big)$ is the geodesic interpolation (or "scaling") of the rotation $x \in SO(3)$ from the identity. $H_\theta(\cdot)[j]$ is a neural network that denoises the orientation and outputs the denoised orientation matrix. The training objective simply minimizes the difference between the real and predicted orientation matrices:

$$L_{\text{ori}}^t = \mathbb{E}\left[ \frac{1}{m} \sum_j \|(O_j^0)^\top \tilde{O}_j^{t-1} - I\|^2 \right] \tag{12}$$

## 4.2 Trajectory Balance Objective

In addition to the denoising objectives, we aim to optimize the binding energy of the generated CDR with respect to the antigen and antibody. However, binding energy can only be computed for the final CDR after a complete denoising process, making it a sparse reward. To address this, we use the **Trajectory Balance (TB)** objective (Malkin et al., 2022) which propagates rewards back through the diffusion trajectory by enforcing global flow-matching constraints.

We begin by defining a GFlowNet state as the partially denoised CDR at timestep $t$. A CDR is composed of a sequence of amino acids and the transition probability for each amino acid location $j$ is the product of the three independent denoising processes (Equations 13 and 14). We define the GFlowNet edge flow as the transition probability of the entire CDR, which is simply the product of the probability of each location (Equations 15 and 16):

$$q(s_j^t|s_j^{t-1}) = q(d_j^t|d_j^{t-1}) \cdot q(x_j^t|x_j^{t-1}) \cdot q(O_j^t|O_j^{t-1}) \tag{13}$$

$$p(s_j^{t-1}|s_j^t) = p(s_j^{t-1}|S^t, C) = p(d_j^{t-1}|S^t, C) \cdot p(x_j^{t-1}|S^t, C) \cdot p(O_j^{t-1}|S^t, C) \tag{14}$$

$$q(S^t|S^{t-1}) = \prod_{j=l}^{l+m} q(s_j^t|s_j^{t-1}) \quad (15) \qquad\qquad p(S^{t-1}|S^t) = \prod_{j=l}^{l+m} p(s_j^t|s_j^{t-1}) \quad (16)$$

For each data point during a mini-batch update, we uniformly sample a timestep $t$ to compute $L_{\text{type}}^t$, $L_{\text{pos}}^t$ and $L_{\text{ori}}^t$. However, we require a complete trajectory to enforce TB. Therefore, we compute all forward $q(S_{t-1}|S_t)$ and backward probabilities $p(S_t|S_{t-1}; \theta)$ for $t$ in $(0, T)$. Following Kim et al. (2024), we precompute the reward $\mathcal{R}(S^0) = \exp(-\alpha \cdot \text{BindingEnergy}(S^0))$ for each CDR $S^0$ in the training dataset and enforce the TB objective:

$$L_{\text{TB}} = \left( \log \frac{Z_\theta \prod_{t=0}^T p(S^t|S^{t-1}; \theta)}{\mathcal{R}(S^0) \prod_{t=0}^T q(S^{t-1}|S^t)} \right)^2 \tag{17}$$

where $Z_\theta$ is the estimated initial state flow and $R(x)$ is the binding energy reward. Therefore, the overall training objective combines the denoising losses and the TB objective:

$$L = \mathbb{E}_{t \sim \text{Uniform}(1, \ldots, T)} \left[ L_{\text{type}}^t + L_{\text{pos}}^t + L_{\text{ori}}^t \right] + w \cdot L_{\text{TB}} \tag{18}$$

where $w$ is a scaling factor for balancing the denoising objectives computed per diffusion step and the TB objective calculated over the entire trajectory.

### 4.3 SAMPLING ALGORITHM

For sequence-structure co-design, we construct $S^T$ by sampling amino acid types for each position from the distribution $d_j^T \sim \text{Uniform}(20)$, CDR positions from the standard normal distribution: $x_j^T \sim \mathcal{N}(0, I_3)$, and orientations from the uniform distribution over SO(3): $O_j^T \sim \text{Uniform}(\text{SO}(3))$. AbFlowNet iteratively denoises the sequence and structures following the standard diffusion process until $t = 0$. Upon generating the amino acid sequence and the structure of the backbone, we optimize the side-chain angles using PyRosetta `PackRotamersMover` (Chaudhury et al., 2010).

Crucially, AbFlowNet applies the GFlowNet balance objective solely during training, not during inference. This approach enables AbFlowNet to operate as a standard diffusion model at sampling time, without requiring energy calculations via the `Rosetta InterfaceAnalyzer`.

## 5 EXPERIMENTS

**Dataset Curation**   We use the Structural Antibody Database (SAbDab) (Dunbar et al., 2014) as the training dataset. We first remove structures whose resolution is less than 4Å and discard antibodies targeting non-protein antigens (Luo et al., 2022). We cluster antibodies in the database according to CDR-H3 sequences at 50% sequence identity using MMSeq2 (Steinegger and Söding, 2017). Our final training dataset contains 9410 antigen-antibody complexes. We evaluate sequence-structure codesign on RAbD test dataset, consisting of 60 diverse antibody-antigen complexes Adolf-Bryfogle et al. (2018b). We also evaluate on the test set proposed by DiffAb (Luo et al., 2022) which contains 19 complexes with antigens from several well-known pathogens including SARS-CoV-2, MERS, influenza, and so on. For both test sets, we strictly remove the overlap between the training set and the testing sets using a CDR-H3 sequence identity threshold of 50%.

**Metrics**   We use standard metrics to evaluate designed antibodies (Adolf-Bryfogle et al., 2018b; Luo et al., 2022; Zhu et al., 2024), namely, (1) **AAR**: the amino acid recovery rate measured by the sequence identity between the reference CDR sequences and the generated sequences, (2) **RMSD**: the $C_\alpha$ root-mean-square deviation (RMSD) between the generated structure and the original structure, and (3) **IMP**: the percentage of designed CDRs with lower (better) binding energy ($\Delta G$) than the original CDR. The binding energy is calculated by `InterfaceAnalyzer` in the Rosetta software package (Adolf-Bryfogle et al., 2018a; Chaudhury et al., 2010). Diffusion models are capable of generating diverse data points from the target distribution by randomly sampling from the initial distribution (Ho et al., 2020; Leach et al., 2022). This is especially advantageous in CDR design where we can generate multiple candidates CDRs *in silico* and select only the most promising CDR for *in vitro* validation according to some desirable property. To this end, the authors of AbDPO (Zhou et al., 2024) proposed: (1) **Top-1 CDR $E_{\text{total}}$**: total energy of the whole designed CDR (kcal/mol) of the *best CDR* out of $N$; (2) **Top-1 CDR-Ag $\Delta G$** : the difference in total energy between the bound state and the unbound state of that CDR and antigen. Following AbDPO, we generate $N$ CDRs for each antigen-antibody complex in the RAbD test dataset and choose the best CDR ranked by $E_{\text{total}} + \Delta G$.

**Model Architecture and Hyperparameters**   We use the transformer-based parametrization defined in Luo et al. (2022) to encode antigen-antibody complex $C$ and conditionally generate $d_j^t$, $x_j^t$ and $O_j^t$. We add a learnable parameter to predict a $Z_\theta$ which is learned solely through backpropagation since $Z_\theta$ global estimation of the initial state's flow independent of individual training samples. Following DiffAb (Luo et al., 2022), we train both DiffAb and AbFlowNet for $200,000$ steps using Adam optimizer (Kingma and Ba, 2014) with learning rate $1e{-}6$. For AbFlowNet, computing TB loss requires sampling full trajectories which is computationally expensive ($\sim 20$ seconds per step). Therefore, we train first $195,000$ steps without TB loss and set TB loss weight $w = 5e{-}6$ for the final 5000 steps. We present details about parameter sweep over $w$ in Appendix B.2 and discuss the effect of training longer in Appendix B.3. Results for methods using sampling budget of $N = 2,528$ were taken from Zhou et al. (2024). We use $N = 100$ when evaluating AbFlowNet for computational efficiency.

## 6 RESULTS

In Section 6.1, we show that AbFlowNet significantly improves Top-1 energy-based metrics and is comparable to AbDPO (Zhou et al., 2024), a far more computationally expensive method. In Section 6.2 we demonstrate that

AbFlowNet's joint optimization improves upon the base diffusion model across all metrics for the same number of training steps. Finally, in Section 6.3, we highlight a qualitative example of the CDR-H3 designed for `PDB 5MES` by DiffAb and AbFlowNet.

## 6.1 *De novo* CDR-H3 design using Top-1 Energy Metrics

Table 1: **Top-1** CDR $E_{\text{total}}$ and CDR-Ag $\Delta G$ (kcal/mol) for *de novo* CDR-H3 design. ($\downarrow$) indicates lower is better. We also show the percentage reduction over DiffAb with the same sampling budget. AbX and AbNovo report Mean CDR $E_{\text{total}}$ and CDR-Ag $\Delta G$ instead of Top-1.

| Methods | Sampling Budget | CDR $E_{\text{total}}$ ($\downarrow$) (with $\Delta$) | CDR-Ag $\Delta G$ ($\downarrow$) (with $\Delta$) | Test Set Used in Sampling | # Param. (M) |
|---|---|---|---|---|---|
| Reference | | 4.52 | -13.72 | — | — |
| **GNN Baselines** | | | | | |
| HERN (Jin et al., 2022) | 2,528 | 7594.94 | 1159.34 | — | 5 |
| MEAN (Kong et al., 2022) | 2,528 | 3113.70 | 114.98 | — | - |
| dyMEAN[1] (Kong et al., 2023) | 2,528 | 15025.67 | 2391.00 | — | - |
| dyMEAN[2] | 2,528 | 3234.30 | 1619.24 | — | - |
| **Diffusion-Based** | | | | | |
| DiffAb (Luo et al., 2022) | 2,528 | 211.00 | 9.54 | — | 4 |
| AbDPO (Zhou et al., 2024)[†] | 2,528 | 162.75 ($\downarrow$23.4%) | **-4.85 ($\downarrow$61.9%)** | Yes | 4 |
| DiffAb | 100 | 480.25 | 11.20 | — | 4 |
| AbFlowNet (Ours) | 100 | **362.03 ($\downarrow$24.8%)** | 1.71 ($\downarrow$38.1%) | **No** | 4 |
| Reference | | — | -19.41 | — | — |
| DiffAb | 128 | — | -0.96 | No | 4 |
| AbX(Zhu et al., 2024) | 128 | — | 4.79 ($\uparrow$31.2%) | No | 12 (+ ESM-3B) |
| AbNovo(Ren et al., 2025) | 128 | — | -12.05 ($\downarrow$60.1%) | No | 12 (+ ESM-3B) |

[†]AbDPO is not open-sourced and cannot be independently reproduced. Results are shown for reference only. Improvements for AbDPO and AbFlowNet are relative to DiffAb under the same sampling budget.

The H3 region is especially difficult for all models to generate because the H3 loop undergoes independent mutation before joining the rest of the antibody sequence (Graves et al., 2020), introducing variability and significantly affecting the structure and function of the antibody.

Table 1 shows that **AbFlowNet is competitive with AbDPO without relying on test-set structures**. Specifically, AbFlowNet significantly outperforms DiffAb at $N = 100$, achieving performance gains comparable to those reported by AbDPO (Zhou et al., 2024), despite not using test-set complexes to generate preference datasets. To evaluate relative improvement, we apply the formula (METHOD - BASELINE) / (REFERENCE - BASELINE), which normalizes performance gains with respect to the baseline and reference. We further validate our findings on the DiffAb test benchmark (Luo et al., 2022), which includes 19 complexes with antigens from SARS-CoV-2, MERS, and influenza; full results are provided in Appendix Table 5.

Compared to AbDPO (Zhou et al., 2024), which is an online RL method that post-trains a DiffAb model to optimize CDR binding energy, AbFlowNet offers several advantages. AbDPO samples 10,122 CDRs per test complex to construct a preference dataset, and updates the model via Direct Preference Optimization (DPO) (Rafailov et al., 2024). This approach has two main limitations: (1) sampling at this scale and computing binding energies is computationally expensive, and (2) using test-set antibody–antigen complexes during RL introduces potential bias. In contrast, AbFlowNet relies solely on precomputed binding energies from the 9,410 training examples, eliminating the need for expensive sampling and reward computation during optimization.

Finally, we note that both CDR $E_{\text{total}}$ and CDR-Ag $\Delta G$ are *Top-1* metrics that select the best-scoring sample among $N$ generated CDR-H3s. These metrics are inherently sensitive to the sampling budget. For example, DiffAb's $E_{\text{total}}$ improves significantly from 480.25 kcal/mol at $N = 100$ to 211.00 kcal/mol at $N = 2,528$. While our evaluation of AbFlowNet uses a modest budget of 100 samples for efficiency, increasing $N$ would likely yield even better results. This suggests that AbFlowNet's performance could scale further with additional samples—without relying on test-set structures or incurring the computational cost of online reward evaluation.

## 6.2 Joint Optimization Outperforms Diffusion-only Baseline

While CDR $E_{\text{total}}$ and CDR-Ag $\Delta G$ measure only binding energy, we evaluate three metrics that compare the generated CDRs to the reference CDR in the dataset. We find that AbFlowNet outperforms DiffAb in all three metrics: $+3.06\%$ in AAR, $+20.40\%$ in RMSD and $+3.60\%$ in IMP when averaged over all 6 CDR regions. We include detailed results in Appendix A. For the CDR-L3 chain, in particular, the RMSD achieved by AbFlowNet is considerably lower than those of other methods. This shows that our Diffusion-GFlowNet Joint Optimization framework maintains strong reconstruction performance, whereas DPO-based post-training RL reduces it. AbDPO, which post-trains only on binding energy, does not preserve the training distribution as well and leads to a 9.96% reduction in average AAR and an increase in RMSD by 0.14 Å. Therefore, AbFlowNet improves binding energy without sacrificing structural accuracy. We find consistent improvements in most CDR regions when using the test set proposed by DiffAb (Luo et al., 2022), shown in Appendix 4.

## 6.3 Qualitative Example

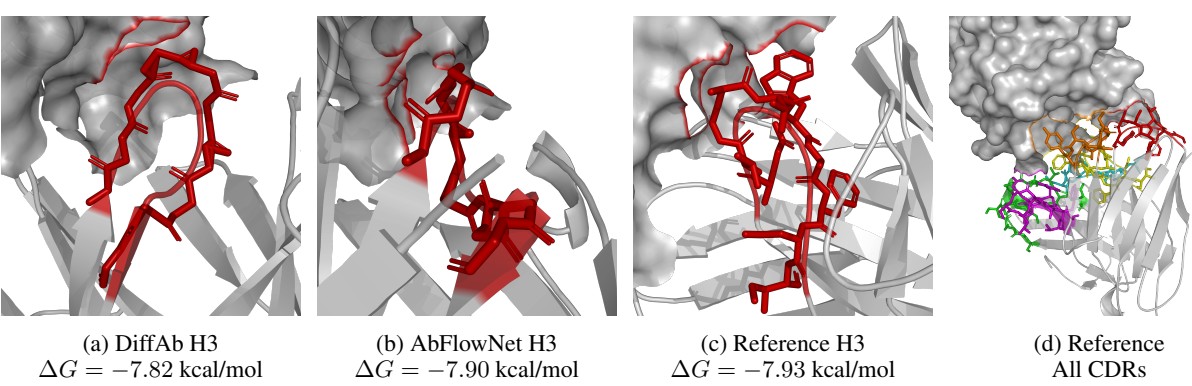

(a) DiffAb H3
$\Delta G = -7.82$ kcal/mol

(b) AbFlowNet H3
$\Delta G = -7.90$ kcal/mol

(c) Reference H3
$\Delta G = -7.93$ kcal/mol

(d) Reference
All CDRs

Figure 3: *De novo* Generated and Reference CDR-H3s for `5MES` complex. For DiffAb and AbFlowNet, we generated 100 CDRs and selected the one with the highest $\Delta$G.

Figure 3 (d) shows Protein Data Bank entry 5MES is a complex where a chimeric mouse-human Fab antibody fragment chaperon is bound to the Mcl-1 antigen. The antigen, a chimeric human/mouse Mcl-1 homolog is over-expressed in various tumors and prevents tumor cells from undergoing apoptosis. The Fab antibody serves to stabilize the complex, allowing researchers to resolve at at 2.24 Å resolution by X-ray diffraction (Johannes et al., 2017). Figure 3 (a) and (b) show that the CDR-H3 region designed by AbFlowNet establishes tighter adhesion between the antibody and antigen. However, *de novo* generation methods still underperform the reference H3, in which side chains contribute a significant fraction of binding affinity. Indeed, CDR side chains may account for the majority of an antibody's binding affinity and specificity (Peng et al., 2014; Robin et al., 2014). Because diffusion-based generators produce only the backbone, we rely on a side-chain packing algorithm (e.g., PyRosetta PackRotamerMover) to place geometrically plausible side-chain orientations. This key limitation of diffusion models is discussed further in Appendix E.

## 7 Discussion

**On the Use of Rosetta InterfaceAnalyzer** AbFlowNet does not strictly require the use of the energy estimators such `PyRosetta InterfaceAnalyzer` (Chaudhury et al., 2010) and could in principle work with *only* the reliable measurements from *in vitro* experiments. This is particularly relevant because prior works have raised concerns about the accuracy of energy estimators (Vreven et al., 2012; Chaves et al., 2023; Conti et al., 2022). In contrast to RL-based approaches such as AbDPO (Zhou et al., 2024), AlignAb (Wen et al., 2024) and AbNovo (Ren et al., 2025), which require energy estimates for thousands of potentially implausible de novo CDRs, AbFlowNet needs energies only for existing reference CDRs. However, of the 9410 complexes in the SAbDab database, only 736 have experimental affinity data available. As such, we used `InterfaceAnalyzer` to estimate the energies for our training set. We opted for `InterfaceAnalyzer` to maintain parity with existing baselines (Luo et al., 2022; Zhou et al., 2024; Wen et al., 2024).

Experimental affinities are typically measured using label-free biophysical techniques such as isothermal titration calorimetry (Boudker and Oh, 2015), surface plasmon resonance (Hearty et al., 2012; Murali et al., 2022) or bio-layer interferometry (Abdiche et al., 2008), each with its own advantages and trade-offs. GPU-accelerated programs such as `OpenMM Yank` (Rizzi et al., 2020) are more accurate but require hours to process a single complex. AbFlowNet

makes it feasible to augment experimental data with GPU-accelerated simulations because these simulations need to be computed only once for authentic CDRs, rather than iteratively for synthetic CDRs.

**Comparison with State-of-the-art methods**   Both the DPO-trained AbDPO (Zhou et al., 2024) and our diffusion-GFlowNet optimized AbFlowNet are based on the same DiffAb (Luo et al., 2022) model as the diffusion backbone, which has 4M parameters, with the key difference being the training framework. While AbFlowNet improves markedly over DiffAb and matches the performance of AbDPO, state-of-the-art methods for CDR design rely on large, billion-scale Protein Language Models to guide the diffusion process while also having a much larger diffusion backbone. This class of models that use Protein LMs, including AbX (Zhu et al., 2024), AbNovo (Ren et al., 2025), and IgGM (Wang et al., 2024), greatly outperforms DiffAb and the models based on it.

In principle, our Diffusion-GFlowNet framework should be compatible with any baseline diffusion model. We selected DiffAb as the backbone for AbFlowNet since we require a lightweight model to sample a complex denoising trajectory and backpropagate through it (We discuss the efficiency considerations of implementing Diffusion-GFlowNet fusion in Appendix F). We attempted pilot experiments by pruning the protein LM component in AbX; however, this caused a complete performance collapse, and the training code for AbX was not publicly available.

## 8   Conclusion

We presented AbFlowNet, a novel framework integrating Diffusion Models and GFlowNets for antibody CDR design. AbFlowNet directly incorporates binding energy signals throughout training, jointly optimizing sequence/structure generation and binding affinity. This approach avoids the trade-offs seen in RL-based methods such as strong reliance on *in silico* binding energy estimation and usage of test set data. AbFlowNet outperforms its base diffusion model (DiffAb) in all metrics and is competitive with expensive RL approaches while only using precomputed rewards.

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

# Technical Appendices and Supplementary Material

## A    PERFORMANCE ON REFERENCE CDR RECONSTRUCTION

Table 2: Evaluation of the generated antibody CDRs (sequence-structure co-design) on the RAbD test dataset (60 sequences) using AAR (%), RMSD (Å) and IMP (%) metrics.

| CDR | Method | AAR↑ | RMSD↓ | IMP↑ | CDR | Method | AAR ↑ | RMSD ↓ | IMP ↑ |
|-----|--------|------|-------|------|-----|--------|-------|--------|-------|
| H1 | Diffab | **64.23%** | 1.153Å | 69.27% | L1 | DiffAb | 53.69% | 1.153Å | 55.51% |
|    | AbFlowNet | 63.49% | **0.974Å** | **73.66%** |    | AbFlowNet | **55.62%** | **0.974Å** | **56.83%** |
| H2 | Diffab | 35.87% | 1.095Å | 46.79% | L2 | DiffAb | 50.46% | 0.795Å | 68.78% |
|    | AbFlowNet | **38.06%** | **0.848Å** | **60.07%** |    | AbFlowNet | **54.09%** | **0.782Å** | **70.64%** |
| H3 | Diffab | 24.34% | 3.236Å | **14.38%** | L3 | DiffAb | **44.87%** | 3.840Å | **36.98%** |
|    | AbFlowNet | **25.08%** | **3.194Å** | 12.65% |    | AbFlowNet | 44.68% | **1.310Å** | 34.70% |

Table 2 shows the performance of AbFlowNet and the baseline diffusion model DiffAb on the RaBD dataset. Both models were trained with identical hyperparameters and the same number of gradient updates; the only difference is that AbFlowNet incorporates the TB objective from Eq. 18.

## B    EXPERIMENTAL SETUP DETAILS

### B.1    HARDWARE SPECIFICATIONS AND RUNTIME

We conduct all experiments using a Linux machine with Intel(R) Xeon(R) Silver 4314 CPU with 512GB memory and one NVIDIA RTX A6000 48GB GPU. Training the first $195,000$ steps without the GFlowNet TB objective took $\sim 27$ hours and training the last $5,000$ steps took $\sim 18$ hours. Sampling 100 times for each CDR regions for every complex in the RAbD test dataset took $\sim 12$ hours.

### B.2    BALANCING BETWEEN DIFFUSION AND TRAJECTORY BALANCE OBJECTIVES

Although the method for computing the forward and backward flow of rewards is computationally expensive, the final trajectory balance loss is simply added to the diffusion reconstruction losses, as shown in Eqn. 18. The Trajectory Balance (TB) loss is typically ranges from $10^4$ to $10^6$, while the three diffusion losses have magnitudes between 0 and 1 after 195000 training steps. This necessitates a TB loss weight $w$ to balance between the flow matching and reconstruction objectives. We train and test AbFlowNet with $w$ ranging from $5e - 5$ to $1e - 7$ and find that learning rates between $1e - 5$ and $1e - 6$ are consistently better than the baseline set by DiffAb. Detailed results are shown in Figure 4.

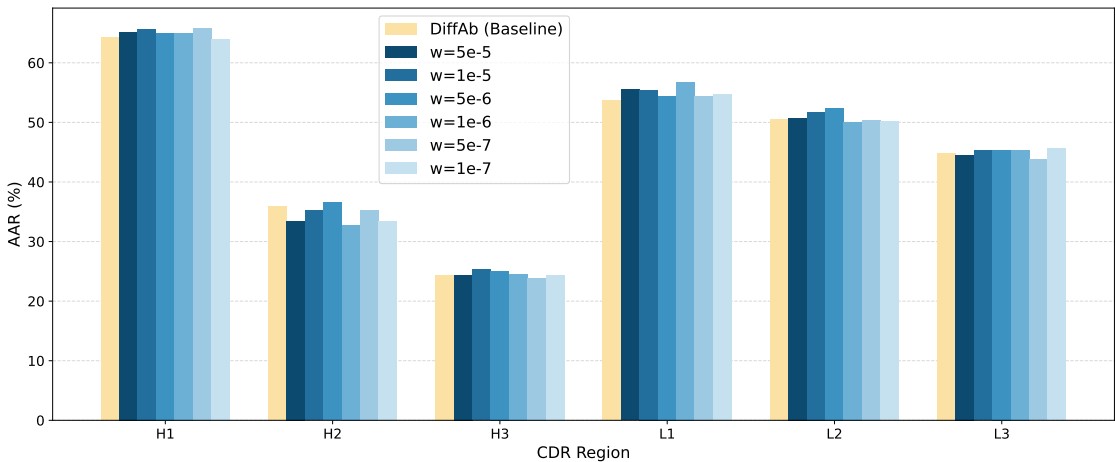

(a) Amino Acid Recovery (ARR) Rate Comparison. Higher is better.

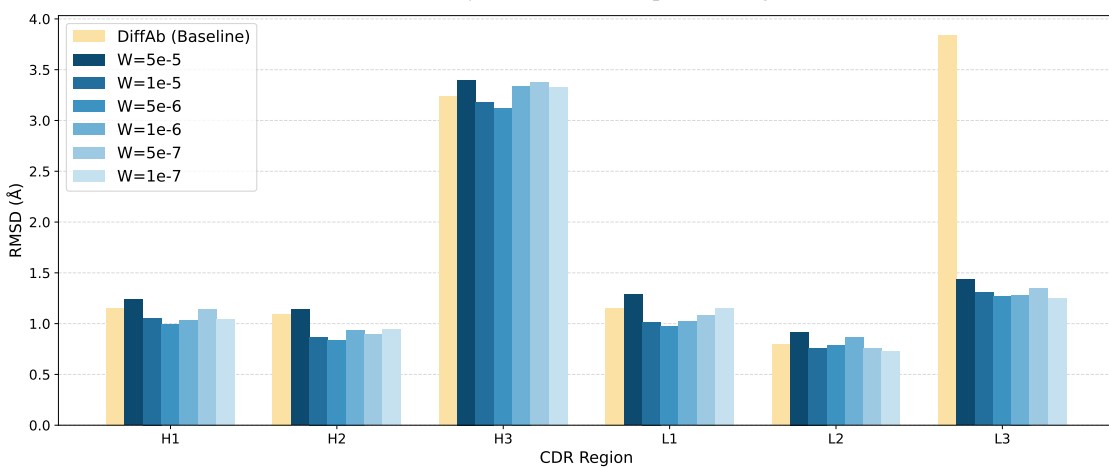

(b) Root Mean Square Deviation (RMSD) Comparison. Lower is better.

Figure 4: Hyperparameter search for TB loss weight $w$ in Eqn. 18 on the RAbD (Adolf-Bryfogle et al., 2018b) dataset. The RMSD of DiffAb on L3 CDR region is significantly worse than AbFlowNet. We repeated the retrained DiffAb using a different seed to confirm this discrepancy (RMSD $4.06$ Å).

## B.3   POST-TRAINING WITH TRAJECTORY BALANCE

So far, we have focused on matching the number of gradient updates between the DiffAb baseline and AbFlowNet to isolate the effect of optimizing binding energy via trajectory balance.

However, training with sparse feedback is often framed as a separate stage after unsupervised learning (Zhou et al., 2024; Zhang et al., 2024b). We test this setup by first training the diffusion model on only the reconstruction objectives for 200K steps and a further 10K steps with the weighted TB loss enabled ($w = 5e - 6$). We find that training beyond 200K steps with reconstruction objectives enables generally tends to overfit the dataset while training with only the TB loss objective harms metrics such as AAR and RMSD, similar to the findings of Zhou et al. (2024). Detailed results are shown in Figure 5.

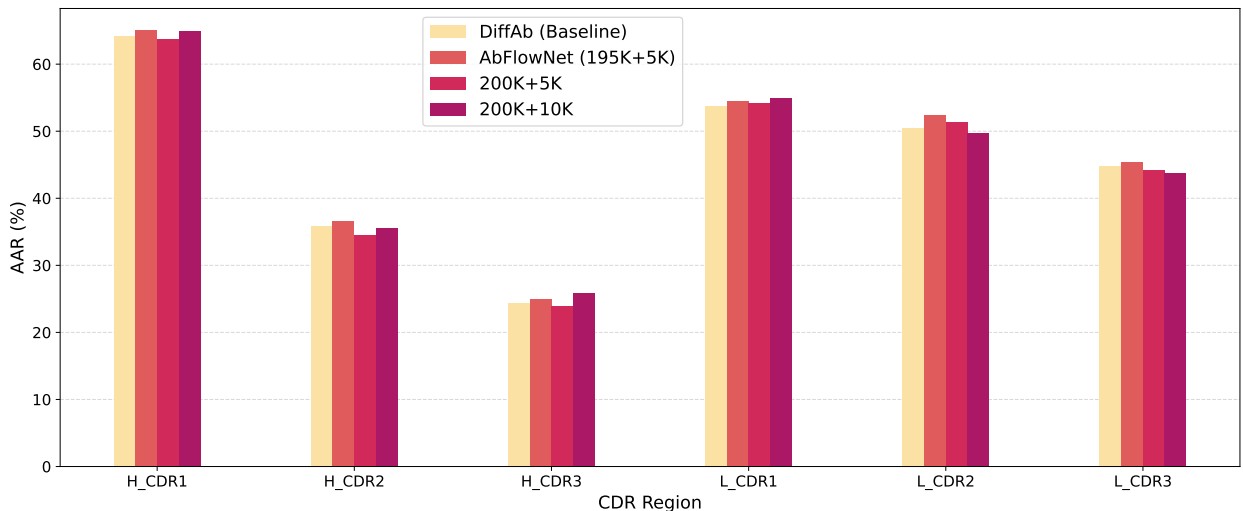

(a) Amino Acid Recovery (ARR) Rate Comparison. Higher is better.

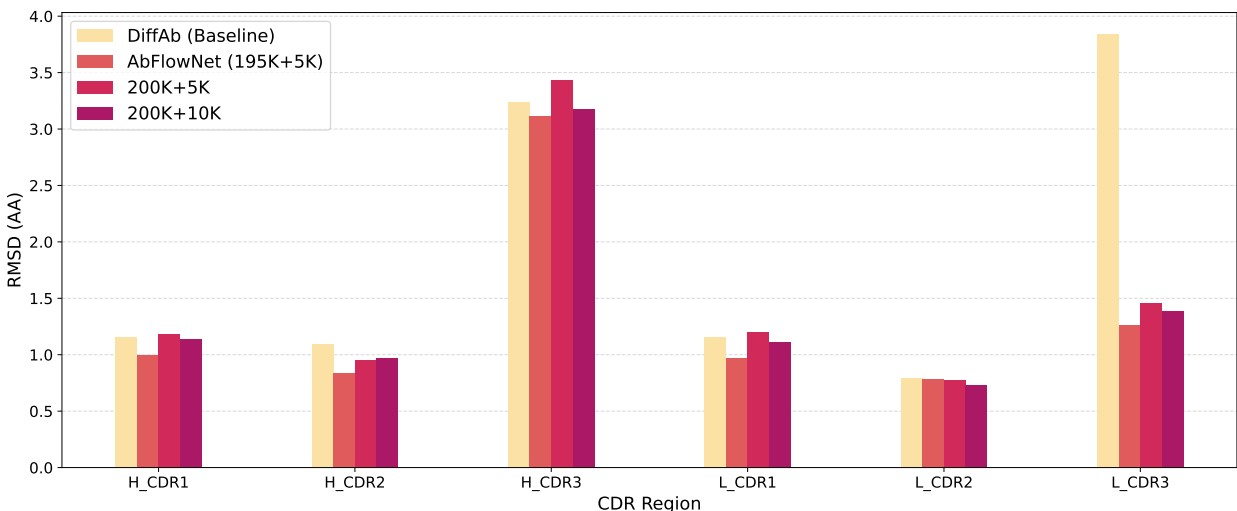

(b) Root Mean Square Deviation (RMSD) Comparison. Lower is better.

Figure 5: Training Diffusion+GFlowNet models with different training steps on the RAbD dataset (Adolf-Bryfogle et al., 2018b). Separating the reconstruction and flow matching steps do not meaningfully improve performance over AbFlowNet.

## C   COMPARISON WITH STATE-OF-THE-ART CDR DESIGN METHODS

Table 3: Evaluation of state-of-the-art CDR design methods on the RAbD test dataset. The DiffAb baseline has been independently benchmarked using different parameters by AbDPO (Zhou et al., 2024), AbNovo (Ren et al., 2025) and IgGM (Wang et al., 2024) which we denote in parenthesis.

| CDR | Method | AAR↑ | RMSD↓ | CDR | Method | AAR↑ | RMSD↓ |
|-----|--------|------|-------|-----|--------|------|-------|
| H1 | DiffAb (Ours) | 64.23 | 1.153 | L1 | DiffAb (Ours) | 53.69 | 1.153 |
|    | AbFlowNet | 63.49 | 0.974 |    | AbFlowNet | 55.62 | 0.974 |
|    | DiffAb (AbDPO) | - | - |    | DiffAb (AbDPO) | - | - |
|    | AbDPO | - | - |    | AbDPO | - | - |
|    | DiffAb (AbNovo) | 70.01 | 0.88 |    | DiffAb (AbNovo) | 61.07 | 0.85 |
|    | AbX | 80.92 | 0.85 |    | AbX | 80.37 | 0.80 |
|    | AbNovo | 85.25 | 0.66 |    | AbNovo | 84.34 | 0.66 |
|    | DiffAb (IgGM) | 63.7 | 0.623 |    | DiffAb (IgGM) | 60.8 | 0.749 |
|    | IgGM | 74.0 | 0.555 |    | IgGM | 75.0 | 0.589 |
| H2 | DiffAb (Ours) | 35.87 | 1.095 | L2 | DiffAb (Ours) | 50.46 | 0.795 |
|    | AbFlowNet | 38.06 | 0.848 |    | AbFlowNet | 54.09 | 0.782 |
|    | DiffAb (AbDPO) | - | - |    | DiffAb (AbDPO) | - | - |
|    | AbDPO | - | - |    | AbDPO | - | - |
|    | DiffAb (AbNovo) | 38.52 | 0.78 |    | DiffAb (AbNovo) | 58.58 | 0.55 |
|    | AbX | 70.73 | 0.76 |    | AbX | 84.53 | 0.45 |
|    | AbNovo | 78.56 | 0.76 |    | AbNovo | 88.25 | 0.32 |
|    | DiffAb (IgGM) | 39.4 | 0.586 |    | DiffAb (IgGM) | 59.9 | 0.466 |
|    | IgGM | 64.4 | 0.486 |    | IgGM | 74.3 | 0.378 |
| H3 | DiffAb (Ours) | 24.34 | 3.236 | L3 | DiffAb (Ours) | 44.86 | 3.840 |
|    | AbFlowNet | 25.08 | 3.194 |    | AbFlowNet | 44.68 | 1.310 |
|    | DiffAb (AbDPO) | 36.42 | 2.34 |    | DiffAb (AbDPO) | - | - |
|    | AbDPO | 26.48 | 2.48 |    | AbDPO | - | - |
|    | DiffAb (AbNovo) | 28.05 | 2.86 |    | DiffAb (AbNovo) | 47.57 | 1.39 |
|    | AbX | 44.18 | 2.50 |    | AbX | 65.89 | 1.21 |
|    | AbNovo | 49.93 | 2.19 |    | AbNovo | 73.88 | 0.86 |
|    | DiffAb (IgGM) | 22.6 | 2.646 |    | DiffAb (IgGM) | 42.4 | 1.017 |
|    | IgGM | 36.0 | 2.131 |    | IgGM | 63.5 | 0.847 |

## D  DIFFAB TEST SET PERFORMANCE

The authors of DiffAb (Luo et al., 2022) proposed a test set consisting of 19 complexes with antigens from several well-known pathogens including SARS-CoV-2, MERS, influenza, and so on. Since these complexes are part of the SAbDab dataset (Dunbar et al., 2014) used for training, we filter our training complexes against the test set using a CDR-H3 sequence identity threshold of 50%. We retrain both DiffAb and AbFlowNet with this new filtered training set.

Table 4: Evaluation of the generated antibody CDRs (sequence-structure co-design) on the DiffAb test dataset (19 sequences).

| CDR | Method | AAR↑ | RMSD↓ | IMP↑ |
|---|---|---|---|---|
| H1 | DiffAb | 68.29% | 1.090Å | 53.96% |
|  | AbFlowNet | **69.41%** | **0.944Å** | **63.50%** |
| H2 | DiffAb | 35.94% | **0.804Å** | **55.60%** |
|  | AbFlowNet | **36.39%** | 0.862Å | 54.38% |
| H3 | DiffAb | 26.53% | **3.183Å** | **12.75%** |
|  | AbFlowNet | **26.66%** | 3.321Å | 8.75% |

| CDR | Method | AAR↑ | RMSD↓ | IMP↑ |
|---|---|---|---|---|
| L1 | DiffAb | 54.40% | **0.960Å** | 62.48% |
|  | AbFlowNet | **55.97%** | 1.019Å | **64.84%** |
| L2 | DiffAb | 42.55% | **0.735Å** | 80.61% |
|  | AbFlowNet | **45.52%** | 0.757Å | **84.47%** |
| L3 | DiffAb | 46.15% | 1.127Å | 37.07% |
|  | AbFlowNet | **46.19%** | 1.180Å | **37.26%** |

Table 5: Summary of Top-1 CDR $E_{total}$ and CDR-Ag $\Delta G$ (kcal/mol) of CDR-H3's designed by DiffAb and AbFlowNet on the DiffAb Test Dataset. ( ↓ ) denotes a smaller number is better.

| Methods | # Samples | CDR $E_{total}$ ( ↓ ) | $+E_{total}$ (%) | CDR-Ag $\Delta G$ ( ↓ ) | $+\Delta G$ (%) | Test Set Used |
|---|---|---|---|---|---|---|
| Reference |  | 1.63 |  | -4.80 |  |  |
| DiffAb | 100 | 26.33 |  | 11.50 |  |  |
| AbFlowNet (Ours) |  | 4.23 | 89.5 | 1.47 | 149.7 | No |

## E  APPROACHES TO DETERMINING SIDE-CHAIN ORIENTATION WITH NEURAL NETWORKS

Most generative methods—including DiffAb, AbDPO, AbFlowNet, AlignAb (Wen et al., 2024), AbX (Zhu et al., 2024), AbNovo (Ren et al., 2025), etc.—generate only the backbone structure. The orientation of the amino-acid chains isn't generated through a diffusion process, since it must follow structural constraints such as avoiding overlaps. Instead, we rely on a side-chain packing algorithm such as `PyRosetta PackRotamerMover` to find the ideal orientation of side-chains. The GNN based dyMEAN (Kong et al., 2023) jointly generates the backbone and side-chain orientations jointly but Table 1 shows that this approach under-performs using `PyRosetta`. There has been notable research into generating only side-chains conditioned on the backbone with diffusion neural networks (Zhang et al., 2023; 2024c) and given the limitations, it is a critical direction of future research.

## F    EFFICIENCY CONSIDERATIONS OF DIFFUSION-GFLOWNET FUSION

Computing the Trajectory Balance (TB) objective requires complete trajectories, i.e., sampling from the initial Gaussian state to the final denoised CDR conditioned on the antigen-antibody complex. Following (Luo et al., 2022; Zhou et al., 2024; Zhu et al., 2024), we use 100 denoising steps. The wall-clock time for 100 denoising steps with a batch size of 16 is $\sim 20$ seconds which dominates the run-time of training AbFlowNet. Furthermore, we do backward propagation for only one random time step since storing the activations for all 100 steps is computationally infeasible. This deviates from the original TB (Bengio et al., 2021) formulation and hence is only as approximation. Following (Zhang et al., 2024b), we discuss our attempt at using an alternative GFlowNet optimization objective, Detailed Balance, in Appendix G.

## G    DETAILED BALANCE OBJECTIVE

Detailed Balance (DB) (Bengio et al., 2021) is an alternative training objective to the standard flow matching constraint and Trajectory Balance which doesn't require enumerating states or sampling complete trajectories. Rather, DB requires the forward flow from state $s$ to $s'$, $F(s)P_F(s'|s)$ to match the backward flow $F(s')P_B(s|s')$. Concretely the DB objective is

$$F(s)P_F(s'|s) = F(s')P_B(s_{t-1}|s_t) \tag{19}$$

However, the flow of a nonterminal state $s$ is generally not tractable, and hence it is parameterized with a neural network $F_\phi(\cdot)$. The forward and backward transition probabilities of the entire CDR $S^t$ are $p(S^{t-1}|S^t) = \prod_{j=l}^{l+m} p(s_j^t|s_j^{t-1})$ and $q(S^t|S^{t-1}) = \prod_{j=l}^{l+m} q(s_j^t|s_j^{t-1})$ respectively. Therefore, the final DB objective is:

$$L_{DB}^t = \left( \log \frac{F_\phi(S^t)p(S^t|S^{t-1};\theta)}{F_\phi(S^{t-1})q(S^{t-1}|S^t)} \right)^2 \tag{20}$$

**Pilot Attempt Using DB Objective**    In the GFlowNet framework, there are three equivalent optimization objectives: Flow Matching (FM), Detailed Balance (DB) and Trajectory Balance (TB) - each with their own tradeoffs. Flow Matching requires enumerating states and enforcing parity between incoming and outgoing flow. FM is not applicable since the number of states in diffusion models is infinite. We attempt using DB which only requires computing the forward and backward flow between two states and enforcing parity.

Similar to optimizing the TB objective outlined in Section 4.2, we uniformly sample a timestep $t$ to compute $L_{type}^t$, $L_{pos}^t$ and $L_{ori}^t$. Since we require adjacent state pairs to compute DB, we do a single step of denoising to obtain $S^{t-1}$ from $S^t$. To enforce the DB objective 19, we must compute $F(S^t)$ and $F(S^{t-1})$. However, intermediate states $S^{t-1}$ and $S^t$ are noisy and therefore are not appropriate to be evaluated by a reward function, which would give noisy results. Following Zhang et al. (2024b), we define the linearization.

$$F_\phi(S^t) = \tilde{F}_\phi(S^t)R(\hat{S}^0) = \tilde{F}_\phi(S^t)R(FullDenoise_\theta(S^t)) \tag{21}$$

where $\tilde{F}_\phi()$ is a scalar function that scales the reward of the estimated fully denoised state. Being a diffusion model, we can fully denoise any noisy state albeit with a sacrifice in quality.

**Key Bottlenecks**    At this stage, we run into the key issue that precludes the use of DB in CDR design. We need to compute the energy of the designed CDR using a tool such as `InterfaceAnalyzer` in the `Rosetta` (Chaudhury et al., 2010) software package. `InterfaceAnalyzer` requires the designed antigen-antibody structure to be complete with side-chains. However, diffusion models generally generate only the backbone and rely on a search-based side-chain packing algorithms such as `PackRotamerMover`. Both `PackRotamerMover` and `InterfaceAnalyzer` are CPU-based utilities and it takes 10.81 seconds to process a single CDR .pdb file. Determining the energy for the two states for each item in the mini-batch (16 in our experiments) requires $\sim 93$ seconds even when parallelized over a 32+ core machine, including multiple data migration costs between the GPU, CPU and disk. This is in contrast to the millisecond-scale time required for the forward and backward passes.

Therefore, the training runtime is dominated by the time it takes to compute the binding energy reward and training becomes infeasible.

**Neural Surrogate for Rosetta's InterfaceAnalyzer**    To the best of our knowledge, a neural network alternative to `PackRotamerMover` and `InterfaceAnalyzer` does not exist. We tried to train a transformer-based neural

network to simulate the function of `InterfaceAnalyzer` directly from the output of the diffusion model. However, this neural network had very low agreement with the `InterfaceAnalyzer` tool (Pearson's coefficient 0.21), which itself is unreliable (Vreven et al., 2012). This is expected since side-chains play a central role in determining binding affinity (Polonelli et al., 2008). Another drawback of the DB objective is the need to compute binding energy of generated CDRs which are not guaranteed to be geometrically plausible.

In light of our findings, we finally committed to Trajectory Balance as the only feasible objective for training AbFlowNet despite the need to sample full trajectories.

## H  LIMITATION

1) The Trajectory Balance objective requires fully generating a CDR, which in our setup requires 100 forward passes with the neural network for each gradient update. 2) As the *in vitro* affinity data for all training complexes is not available and for fair comparison with existing methods, we used `Pyrosetta InterfaceAnalyzer` which is an unreliable estimator of binding energy. 3) Due to compute constraints, we sampled 100 CDRs per complex in Table 6.1. Sampling at higher rates would potentially increase Top-1 metrics.

# I DISTRIBUTION OF CDR ENERGY METRICS

Table 6: Mean, standard deviation and Minimum of Energy Metrics of CDRs generated by AbFlowNet on the RAbD test dataset. † denotes complexes that could not be processed with PyRosetta FastRelax.

| Complex | Energy of Complex $E_{complex}$ (↓) | | | Binding Energy CDR-Ag $\Delta G$ (↓) | | | Total CDR Energy $E_{total}$ (↓) | | |
|---|---|---|---|---|---|---|---|---|---|
| | Min | Mean ± Std | Ref | Min | Mean ± Std | Ref | Min | Mean ± Std | Ref |
| 1a14_H_L_N | 465.24 | 739.27 ± 349.05 | 1829.81 | -23.46 | 5.76 ± 154.80 | 185.48 | 18.46 | 151.25 ± 168.06 | 56.52 |
| 1a2y_B_A_C | -695.23 | -643.06 ± 86.15 | -412.00 | -20.92 | -15.31 ± 2.99 | 17.32 | -9.88 | 10.63 ± 12.45 | -13.09 |
| 1fe8_H_L_A † | 66.58 | 2244.82 ± 1279.23 | -300.06 | 151.96 | 1998.45 ± 1199.38 | 62.53 | 212.53 | 1444.82 ± 731.12 | 29.13 |
| 1ic7_H_L_Y † | 94.59 | 920.13 ± 702.52 | 29.49 | 54.26 | 250.78 ± 308.75 | 75.85 | 54.36 | 541.44 ± 397.55 | 17.72 |
| 1iqd_B_A_C | -591.29 | -464.97 ± 75.50 | -60.88 | -37.13 | 16.03 ± 35.73 | 206.79 | -1.97 | 33.88 ± 21.23 | 41.34 |
| 1n8z_B_A_C | 64.03 | 258.41 ± 166.25 | 2867.82 | 10.21 | 82.56 ± 82.50 | 188.81 | 8.13 | 95.22 ± 72.46 | 34.61 |
| 1ncb_H_L_N | 567.36 | 1235.35 ± 781.95 | 1602.17 | -4.30 | 128.75 ± 273.55 | 188.93 | 27.20 | 241.38 ± 337.78 | 31.78 |
| 1osp_H_L_O | -396.07 | -69.62 ± 279.88 | -10.07 | -18.15 | 61.66 ± 183.67 | 75.33 | 20.21 | 149.73 ± 135.03 | -2.64 |
| 1uj3_B_A_C | -611.02 | -250.98 ± 309.79 | -214.20 | -24.65 | 26.25 ± 58.20 | 54.90 | -1.91 | 36.66 ± 38.08 | -4.91 |
| 2adf_H_L_A | -519.56 | -283.61 ± 196.12 | -591.05 | -16.85 | 54.62 ± 127.52 | 25.70 | 10.47 | 66.50 ± 34.71 | -12.68 |
| 2b2x_H_L_A | -314.15 | -26.41 ± 231.04 | 375.35 | -17.12 | 24.69 ± 75.21 | 117.86 | 11.17 | 70.64 ± 60.82 | 6.81 |
| 2cmr_H_L_A | -581.82 | -355.59 ± 240.43 | 131.70 | -18.15 | 90.94 ± 169.18 | 106.36 | 6.15 | 80.40 ± 77.97 | -3.14 |
| 2dd8_H_L_S | -17.64 | 230.40 ± 155.19 | 424.31 | -93.32 | -8.70 ± 20.69 | 181.95 | -1.80 | 38.35 ± 36.71 | 74.37 |
| 2ghw_B_b_A | -501.10 | -373.60 ± 151.76 | -124.77 | -33.58 | -11.66 ± 31.11 | 84.61 | -0.74 | 39.36 ± 67.84 | 15.03 |
| 2vxt_H_L_I | -371.27 | 686.03 ± 1004.31 | -426.22 | 26.07 | 599.32 ± 838.56 | 32.43 | 25.65 | 641.44 ± 579.75 | -5.41 |
| 2xqy_G_L_A † | -805.28 | -673.89 ± 217.51 | -61.00 | -20.04 | -2.50 ± 8.44 | 33.51 | -5.38 | 36.07 ± 32.74 | -2.43 |
| 2xwt_A_B_C | -793.86 | -472.31 ± 246.79 | -614.54 | 16.28 | 132.46 ± 60.82 | 21.49 | 2.00 | 66.14 ± 51.55 | -14.67 |
| 2ypv_H_L_A | -854.85 | -705.91 ± 116.26 | -589.38 | -18.71 | 10.92 ± 24.19 | 37.34 | 7.40 | 52.64 ± 39.39 | 5.15 |
| 3bn9_D_C_B | -406.84 | 1233.60 ± 1772.78 | 256.31 | -16.47 | 815.04 ± 1437.08 | 299.53 | 52.33 | 1029.08 ± 985.85 | 133.68 |
| 3cx5_J_K_E | -175.80 | 154.47 ± 320.10 | 265.42 | -14.12 | 5.47 ± 25.08 | 60.79 | 21.83 | 171.98 ± 154.79 | -10.61 |
| 3h3b_c_C_B | -458.08 | -233.14 ± 277.84 | -11.97 | 2.34 | 6.65 ± 4.86 | 41.78 | 11.24 | 96.08 ± 84.23 | 10.90 |
| 3hi6_X_Y_B | -580.80 | -180.53 ± 502.57 | -374.75 | -17.26 | 138.10 ± 403.76 | 49.91 | 20.04 | 181.51 ± 198.83 | 5.80 |
| 3k2u_H_L_A | 2288.98 | 2402.07 ± 83.48 | 2858.45 | -26.46 | 3.46 ± 19.54 | 110.99 | 9.60 | 79.59 ± 58.88 | 18.79 |
| 3l95_B_A_X | -92.20 | 164.80 ± 119.44 | 219.03 | -12.62 | 27.84 ± 49.06 | 149.92 | 12.96 | 85.10 ± 45.78 | 2.93 |
| 3mxw_H_L_A | -483.43 | -339.37 ± 120.16 | -485.02 | -17.19 | -0.88 ± 16.07 | 26.31 | 6.20 | 55.40 ± 36.81 | -1.16 |
| 3nid_H_L_AD | -1627.05 | -1420.20 ± 157.44 | -1232.12 | -22.66 | 30.28 ± 25.74 | -1.90 | 16.14 | 119.02 ± 81.13 | -14.81 |
| 3o2d_H_L_A | -647.48 | -320.46 ± 339.92 | 99.11 | -108.43 | 19.95 ± 87.34 | 116.32 | 15.60 | 156.66 ± 162.21 | 0.16 |
| 3rkd_H_L_A | -559.58 | 546.00 ± 1289.63 | -293.93 | -26.78 | 237.52 ± 414.37 | -18.92 | 27.52 | 541.60 ± 615.68 | -2.53 |
| 3s35_H_L_X | -645.40 | -563.05 ± 80.61 | -123.47 | -20.89 | 14.90 ± 62.73 | 65.90 | -1.80 | 31.35 ± 29.16 | 1.75 |
| 3uzq_A_a_B | -544.58 | 442.73 ± 2213.79 | 27501.00 | -42.78 | -2.51 ± 123.08 | - | 7.26 | 191.34 ± 260.35 | -15.86 |
| 3w9e_A_B_C | -545.74 | -72.71 ± 407.60 | -129.53 | 2.03 | 100.45 ± 240.09 | 88.40 | 36.48 | 210.11 ± 162.17 | -2.35 |
| 4cmh_B_C_A | -836.91 | -441.40 ± 309.48 | -591.38 | -30.57 | 16.98 ± 47.83 | 51.93 | 18.21 | 187.31 ± 134.80 | -10.43 |
| 4dtg_H_L_K | -300.15 | -99.86 ± 192.58 | -125.16 | -6.02 | 18.94 ± 25.45 | 58.17 | 9.78 | 127.47 ± 144.68 | 15.80 |
| 4dvr_H_L_G | -366.66 | -253.46 ± 123.38 | 146.55 | -19.77 | -2.01 ± 38.03 | 58.58 | -8.60 | 40.92 ± 53.03 | -1.64 |
| 4etq_H_L_C | -564.55 | -337.49 ± 177.81 | -161.47 | -0.33 | 115.77 ± 50.70 | 73.30 | -3.46 | 41.51 ± 38.69 | -2.37 |
| 4ffv_D_C_B | -610.13 | -399.67 ± 131.91 | 347.73 | -20.34 | -6.64 ± 13.72 | 90.90 | 6.48 | 31.31 ± 28.74 | 26.73 |
| 4fqj_H_L_A | -191.74 | 1270.81 ± 1615.52 | 329.45 | 2.09 | 662.42 ± 998.74 | 130.07 | 75.53 | 732.24 ± 771.87 | 21.99 |
| 4g6j_H_L_A | -284.25 | -177.17 ± 106.34 | -278.38 | 2.98 | 39.60 ± 32.32 | 97.63 | -4.55 | 43.90 ± 31.31 | 7.59 |
| 4g6m_H_L_A | -727.04 | -621.49 ± 122.25 | -416.44 | -16.21 | 3.93 ± 29.46 | 36.74 | -7.36 | 42.04 ± 41.27 | -7.57 |
| 4h8w_H_L_G | -765.55 | -525.18 ± 167.65 | -500.04 | -22.41 | 62.50 ± 97.84 | 49.32 | 11.28 | 104.24 ± 83.04 | 1.33 |
| 4ki5_E_F_M | -193.51 | 289.45 ± 524.73 | -175.88 | -22.73 | 129.03 ± 232.47 | 45.83 | 22.34 | 211.71 ± 275.09 | -6.51 |
| 4lvn_C_B_A | -650.42 | -148.01 ± 571.38 | -212.13 | -25.83 | 94.06 ± 208.54 | 68.50 | 20.18 | 170.38 ± 233.80 | 29.01 |
| 4ot1_H_L_A | -377.87 | 1924.70 ± 2775.83 | -332.76 | -5.03 | 489.36 ± 1109.65 | 62.94 | 95.39 | 1792.68 ± 2029.26 | -2.69 |
| 4qci_B_A_D | -299.23 | -130.16 ± 123.36 | -76.90 | -17.54 | 53.22 ± 52.44 | 37.63 | 5.22 | 63.86 ± 63.54 | 14.68 |
| 4xnq_B_A_D | 190.39 | 582.77 ± 311.38 | 238.64 | -10.39 | 54.82 ± 71.60 | 34.20 | 41.31 | 271.33 ± 197.50 | -9.83 |
| 4ydk_H_L_G | -754.23 | 1338.02 ± 2353.61 | -617.74 | -25.80 | 841.97 ± 1496.09 | 22.24 | 73.71 | 1361.75 ± 1483.04 | -23.96 |
| 5b8c_B_A_C | -405.83 | -89.10 ± 259.21 | -322.93 | 15.95 | 119.52 ± 80.35 | 90.55 | 20.50 | 120.70 ± 107.06 | 0.10 |
| 5bv7_C_B_A | 196.59 | 920.92 ± 828.95 | 1230.31 | 21.23 | 172.09 ± 217.10 | 113.39 | 49.49 | 447.16 ± 464.02 | 44.74 |
| 5d93_C_B_A † | -479.00 | 805.38 ± 1122.77 | -541.29 | 25.53 | 729.50 ± 854.56 | 22.08 | 29.29 | 860.37 ± 716.65 | -0.84 |
| 5d96_J_I_D | -197.67 | -50.32 ± 191.34 | 295.45 | -15.31 | 23.77 ± 103.76 | 87.44 | 1.94 | 91.77 ± 100.97 | 11.20 |
| 5en2_A_B_C | -681.11 | -257.33 ± 683.81 | -532.16 | -27.48 | 44.84 ± 220.13 | 14.59 | 20.79 | 280.10 ± 378.01 | -12.95 |
| 5f9o_H_L_G | -748.61 | -416.83 ± 315.42 | 67.19 | -39.86 | 13.56 ± 64.36 | 88.09 | 12.70 | 168.53 ± 152.44 | 4.09 |
| 5ggs_A_B_Z | -412.43 | -123.40 ± 249.30 | -137.59 | -12.58 | 52.22 ± 82.59 | 56.20 | 11.03 | 129.07 ± 135.37 | -8.39 |
| 5hi4_H_L_A | -508.21 | -398.51 ± 114.40 | -215.09 | -20.36 | 8.50 ± 96.39 | -24.37 | 6.73 | 55.59 ± 33.81 | 8.09 |
| 5j13_C_B_A | -475.93 | -298.67 ± 151.07 | -365.81 | -18.03 | 1.64 ± 19.68 | 16.54 | 16.60 | 116.51 ± 103.25 | 5.91 |
| 5l6y_H_L_C | -487.69 | -190.69 ± 319.95 | -360.32 | -29.21 | 20.40 ± 68.66 | -3.57 | 21.19 | 164.42 ± 174.97 | 6.56 |
| 5mes_H_L_A | -594.11 | -422.81 ± 94.28 | -156.46 | -7.90 | -0.74 ± 7.36 | -7.93 | -2.23 | 43.10 ± 35.82 | -9.65 |
| 5nuz_A_B_C | -758.42 | -556.64 ± 239.14 | -674.95 | -31.56 | -1.03 ± 36.37 | -11.37 | 11.88 | 87.82 ± 70.17 | -9.43 |

Figure 6: Distribution of sample CDRs generated by AbFlowNet on the RAbD test dataset.

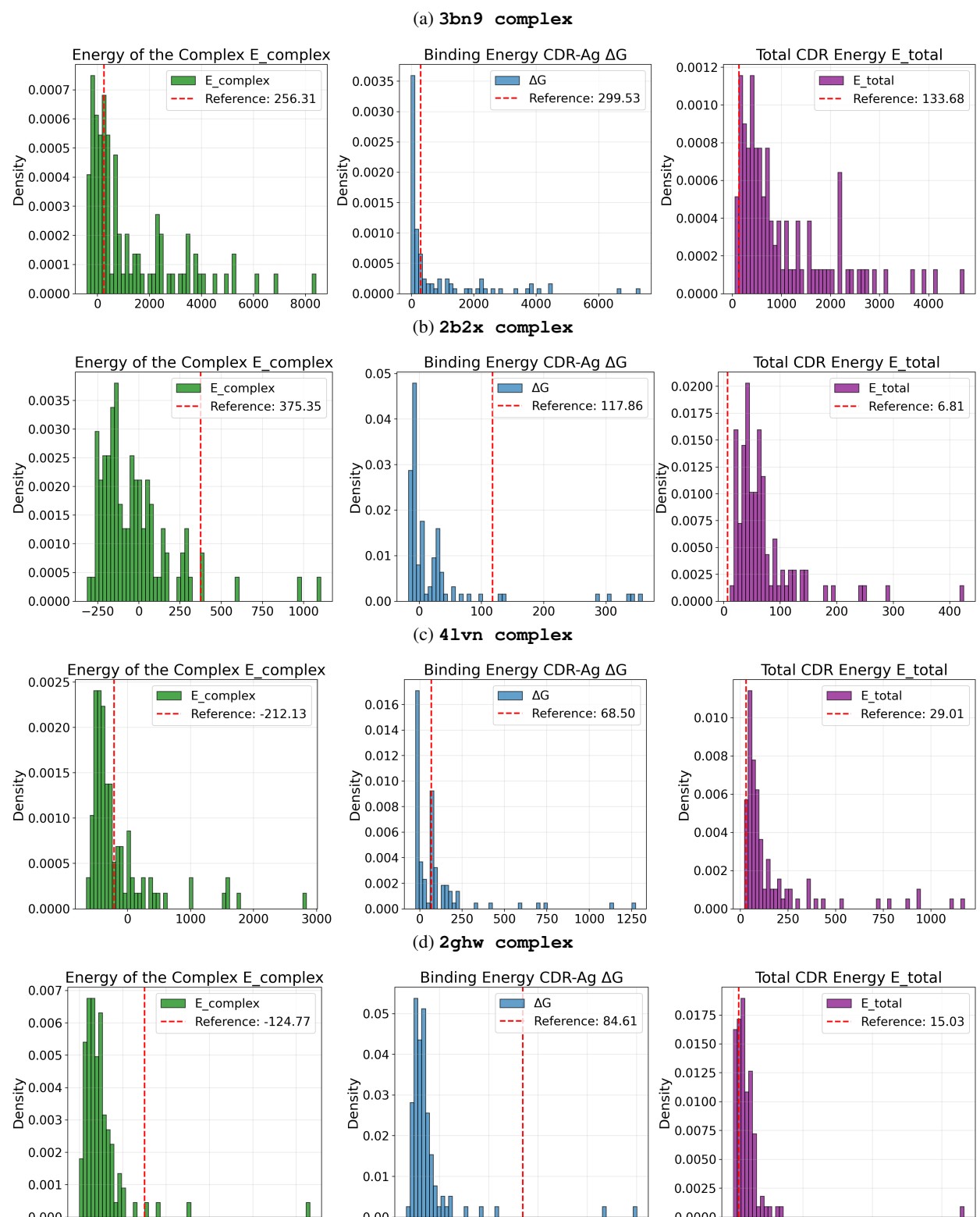

## J    ADDITIONAL QUALITATIVE EXAMPLES AND INTERPRETATION

We visualize the generated CDRs using PyMol (DeLano et al., 2002). The globular structure on the top is the molecular surface of the antigen and the ribbon-like structure at the bottom is the antibody, with the CDR-H3 region colored in. The large arrows represent the start and end of the rest of the antibody sequence that we keep fixed. The antigen surface has grooves and cavities where the CDR region binds to. Omitting salt-bridge and hydrogen bond interactions for simplicity, we color code the hydrophilic and hydrophobic residues of the antigen and the CDR. Hydrophobic antigen residues are colored in orange and hydrophobic CDR residues are colored in red. Likewise, yellow and green represent hydrophilic residues in the antigen and CDR respectively. **Naturally, due to specificity, a stable configuration would have matching regions closer together, orange-red for hydrophobic and yellow-green for hydrophilic. Another aspect of consideration is the Binding Surface Area (BSA) which measures how deeply inset the CDR region is in the antigen.** Specificity, surface area and other factors such as the existence of salt bridges and hydrogen bonds together determine the overall stability and binding energy.

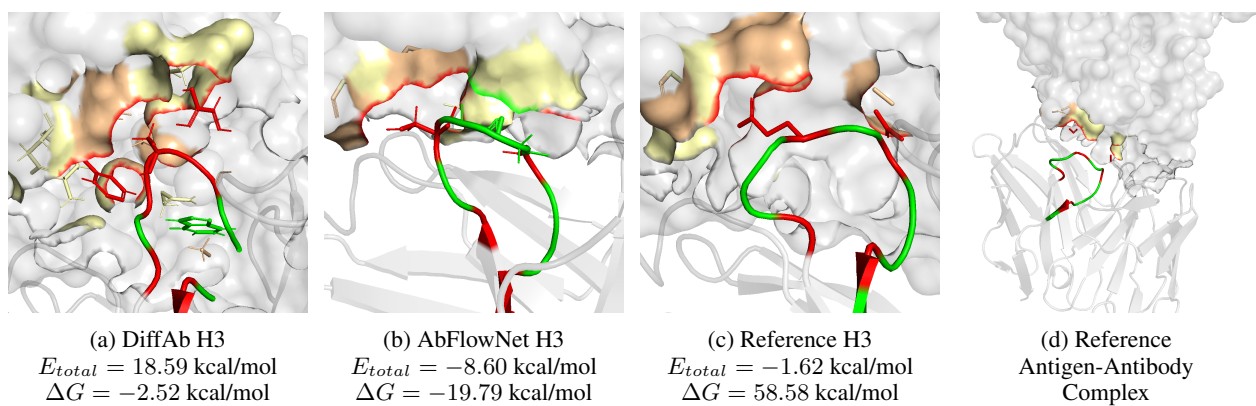

| (a) DiffAb H3 | (b) AbFlowNet H3 | (c) Reference H3 | (d) Reference Antigen-Antibody Complex |
|---|---|---|---|
| $E_{total} = 18.59$ kcal/mol | $E_{total} = -8.60$ kcal/mol | $E_{total} = -1.62$ kcal/mol | |
| $\Delta G = -2.52$ kcal/mol | $\Delta G = -19.79$ kcal/mol | $\Delta G = 58.58$ kcal/mol | |

Figure 7: *De novo* Generated and Reference CDR-H3s for `4dvr` complex. We selected the one with the highest $\Delta G$ out of 100 generations for DiffAb and AbFlownet.

Figure 7 highlights a case where AbFlowNet designs a CDR-H3 region that is embedded deeper in the antigen surface than the reference. The reference CDR-H3 region is further apart from the antigen surface and has higher Binding Energy $\Delta G$ as consequence. Both AbFlowNet and the reference has minor regions of poor specificity where hydrophobic and hydrophilic residues are color together (red-yellow interaction). In contrast, Diffab generates a CDR almost entirely with hydrophobic (red) residues and has large regions of poor specificity.

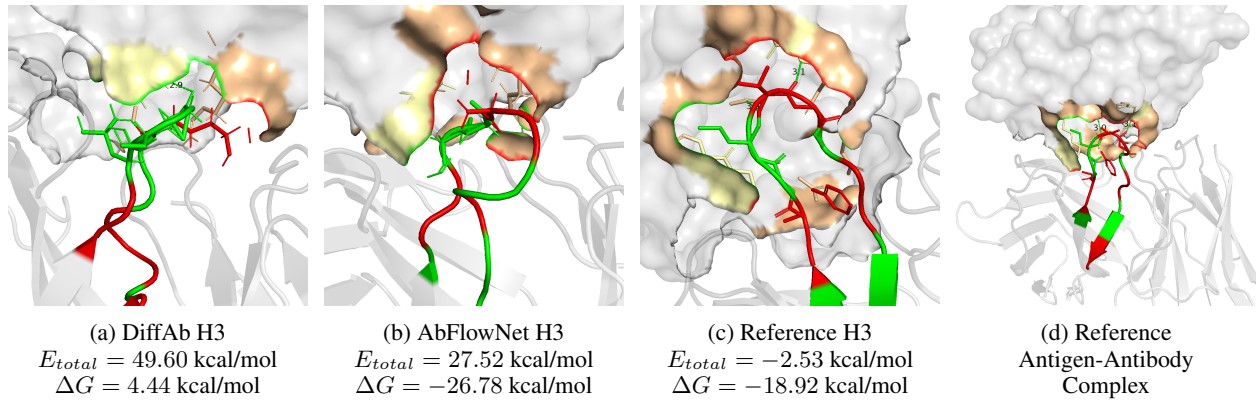

| (a) DiffAb H3 | (b) AbFlowNet H3 | (c) Reference H3 | (d) Reference Antigen-Antibody Complex |
|---|---|---|---|
| $E_{total} = 49.60$ kcal/mol | $E_{total} = 27.52$ kcal/mol | $E_{total} = -2.53$ kcal/mol | |
| $\Delta G = 4.44$ kcal/mol | $\Delta G = -26.78$ kcal/mol | $\Delta G = -18.92$ kcal/mol | |

Figure 8: *De novo* Generated and Reference CDR-H3s for `3rkd` complex.

Figure 8 shows that the reference CDR-H3 for `3rkd` complex is embedded deeply in the antigen. Due to the deep inset, the reference is significantly lower Total CDR Energy $E_{total} = -2.53$ than the generated CDRs. The CDR generated by AbFlowNet is also inset deeply but has a smaller binding surface area. Diffab's CDR-H3 is the furthest from the molecular surface of the antigen. The reference CDR has regions of poor specificity (orange-green interaction) while AbFlowNet shows high specificity.

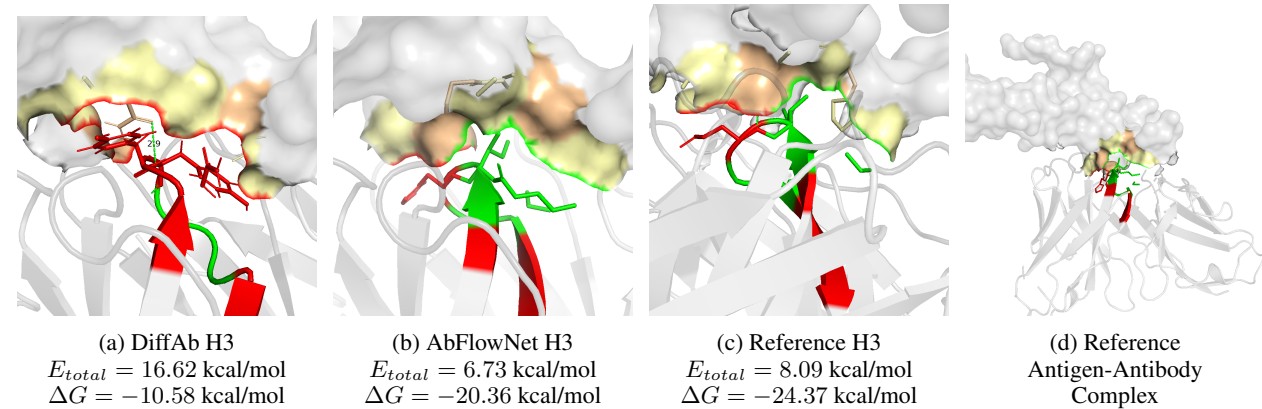

(a) DiffAb H3
$E_{total} = 16.62$ kcal/mol
$\Delta G = -10.58$ kcal/mol

(b) AbFlowNet H3
$E_{total} = 6.73$ kcal/mol
$\Delta G = -20.36$ kcal/mol

(c) Reference H3
$E_{total} = 8.09$ kcal/mol
$\Delta G = -24.37$ kcal/mol

(d) Reference
Antigen-Antibody
Complex

Figure 9: *De novo* Generated and Reference CDR-H3s for `5hi4` complex.

Figure 9 highlights a case where AbFlowNet generates a CDR-H3 very similar to the reference CDR-H3, differing only in one residue and minimal differences in position and orientation. As such $\Delta G$ and $E_{total}$ are very similar. We note that our training set was deduplicated precisely based on $\geq 50\%$ similarity with CDR-H3 region. This is a strong signal that AbFlowNet implicitly learns structures that optimize binding energy due to GFlowNet training. In contrast, similar to Figure 7, DiffAb once again generates a CDR-H3 region with only hydrophobic (red) residues and shows poor specificity (red hydrophobic CDR residues close to yellow hydrophilic antigen surface) despite having a deeper inset than even the reference.

## K EVOLUTIONARY PLAUSIBILITY OF GENERATED CDRS

Protein Language Models (PLM) have been shown to learn evolutionary plausibility of protein sequences (Lin et al., 2022) and have been successfully used in wet-lab applications (Zhang et al., 2025). We use the ESM-2 PLM to extract embeddings of the generated CDRs and compare them to the embeddings of the reference CDR to determine their evolutionary plausiblity. Since the CDR regions are small, we use Locality-Aware Pooling (Hoang and Singh, 2025) to extract the CDR's embedding conditioned on the local antibody chain. Figure 10 shows that generated CDRs sequences strongly cluster together with their corresponding reference sequences, showing that the generated CDRs are evolutionarily plausible and highly similar to their references according to ESM-2.

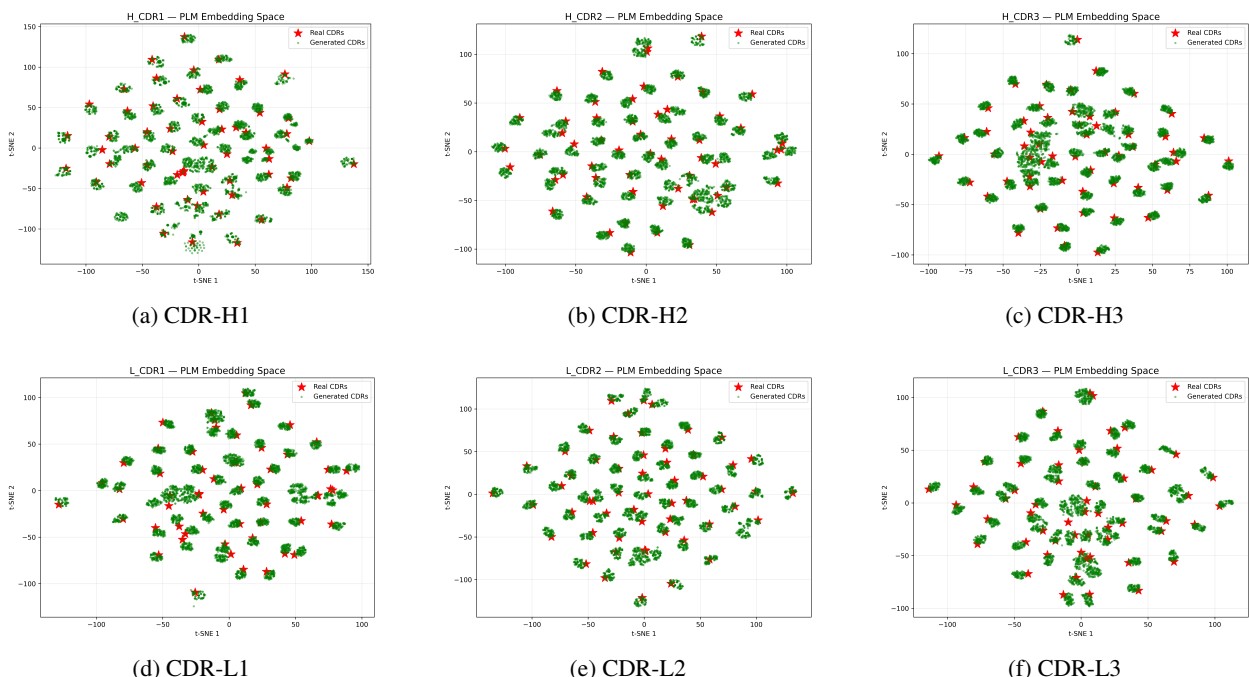

Figure 10: tSNE clustering of ESM-2 embeddings of CDRs generated by AbFlowNet and the reference CDR of the RAbD dataset. The reference CDR is generally in the same cluster as the generated CDRs, showing that these are evolutionarily plausible and possess natural-like sequence properties according to ESM-2.

## L ACKNOWLEDGMENT REGARDING USE OF LARGE LANGUAGE MODELS

We used Large Language Models (LLM) only as a writing aid to rephrase and correct the draft. We did not use LLMs to generate code, experiments or ideation.

