# OpenReview forum: "AbFlowNet: Optimizing Antibody-Antigen Binding Energy via Diffusion-GFlowNet Fusion"
_ICLR.cc/2026/Conference — Submitted to ICLR 2026_

### Official Review · Reviewer_Rcth · 2025-10-25

**Soundness:** 2
**Presentation:** 3
**Contribution:** 2
**Rating:** 4
**Confidence:** 4

**Summary:**

This paper introduces AbFlowNet, a framework that unifies diffusion models and GFlowNets for antibody–antigen design.The core idea is to reinterpret each denoising step in a conditional diffusion process as a trajectory transition within a GFlowNet, where the terminal state corresponds to a fully denoised CDR sequence + backbone, and its reward is proportional to binding energy (computed offline via Rosetta InterfaceAnalyzer). Training uses a Trajectory Balance (TB) objective jointly with standard diffusion reconstruction losses, allowing reward information to propagate along the entire denoising trajectory without requiring online energy evaluation.

**Strengths:**

This work is the first to treat diffusion denoising trajectories as GFlowNet paths, which is an elegant conceptual unification, and seems to achieve good results.

**Weaknesses:**

1. In my experience, the Rosetta energy evaluation is quite cheap and i do not think online RL is an issue. Especially considering the protein design area, the real cost is never the simulation efforts but the wet-lab evaluation. So i think online RL makes more sense

2. The Rosetta evaluation is not that reliable. Does the author have any wet-lab exps?

3. Eq.(18) introduces an extra hyperparameter w. It can be quite tricky to tune this w for different datasets.

4. In fact, there has been work using RL to fine-tune diffusion models by treating denoising trajectories as paths. https://arxiv.org/abs/2305.13301 but i did not see the authors compare it or mention it. RL is closely relateed to GFlowNet so this denoising traj as path contribution is not that strong considering this paper.

5. The authors seem to retrain diffab but there is already diffab checkpoint. Why re-train it?

6. Also, the authors seem to emphasize the same gradient update step for baselines. I do not think that is the real metric we should care about or i think we should not care that metric at all. First, the real metric should be sth like FLOPS, as the gradient step for different model/algo can be different. Second, as i mention before, we should not care about much about computational cost in the protein domain, and it does not matter as much as we did in general ML area (the comp is not large in bio domain or that important).

**Questions:**

See Weakness

---

> ### Author Response · Authors · 2025-11-21
> **Response to Reviewer Rcth (Part 1 of 2)**
>
> Thank you for the constructive feedback.  We appreciate your comments and address each of your concerns below.
>
> 1. **On the Use of Rosetta and Lack of Wet-Lab Experiments**
>
> We agree that Rosetta is an imperfect and noisy proxy for true binding energy. As discussed in the manuscript (L446), AbFlowNet does not require an energy estimator at inference time and could, in principle, be trained entirely with wet-lab binding measurements because rewards are needed only for the fixed training dataset. Unlike online RL approaches, AbFlowNet does not need binding energies for newly generated CDRs, which eliminates the risk of compounding estimator noise during training.
>
> We used Rosetta for two reasons: ***1) Wet-lab measurements are not yet available for all complexes in the SAbDab training set, and 2) Consistent comparison with baselines such as AbDPO, AbX, and AbNovo requires using the same energy estimator they rely on.*** Using a different estimator would make the comparisons inconclusive.
>
> We fully agree that wet-lab validation is the ultimate evaluation for computational antibody design, and we identify this as an important direction for future work.
>
> 2. **Pitfalls of Online RL: Overfitting to Noisy Energy Estimators and Computational Cost**
>
> Previous works such as AbDPO and AbNovo have used DPO-based RL for binding energy optimization. To this end, their respective authors generated thousands of new CDRs and scored them with Rosetta in order to optimize the model. However, Rosetta is only a noisy estimator of the true binding energy, and optimizing based on this unreliable reward signal runs the serious risk of the classical RL issue of **reward hacking**. In fact, for AbDPO, this RL optimization significantly decreased reconstruction-based metrics such as AAR and RMSD, which track biological plausibility against the real reference CDR.
>
> In contrast, our current AbFlowNet pipeline only uses Rosetta estimates on reference CDRs which prevents reward hacking. Furthermore, our joint optimization of diffusion reconstruction and GFlowNet binding energy ensures that the model doesn't trade off binding energy at the cost of biological plausibility.
>
> **Regarding computational cost:** While Rosetta InterfaceAnalyzer is relatively fast for estimating binding energy, we must also carry out a relaxation step before estimating binding energy. Since diffusion models are not physics-aware, the generated side-chains often collide with each other. Tools like Rosetta FastRelax optimize the side-chains to minimize collision by heuristically searching over the exponential search space of side-chain orientations. In our experiments, FastRelax takes an average of 1.48 minutes to optimize the CDR-H3 region of a single complex (6-20 residues). As such, the cost for relaxation and binding energy optimization in online RL is significant. Online RL would also be unable to utilize more reliable estimators such as the GPU-based OpenMM Yank, which can take hours for a single complex. Since AbFlowNet requires only the binding energy of real CDRs, we can forgo the relaxation step altogether and precompute energies with reliable estimators.
>
> 3. **Why Retrain DiffAb?**
>
> The original DiffAb checkpoint was trained on complexes in the RAbD test dataset. The authors of DiffAb curated their own test set with 19 disease-related complexes. Later work such as AbDPO, AbX, and AbNovo have elected to use the 60 complexes of the RAbD test set. To ensure there was no test-set leakage, we rigorously filtered the training set (L313) and retrained DiffAb.
>
> 4. **Interpretation of Gradient Update Steps**
>
> We fully agree that gradient update steps are not a practical measurement of the compute required to train models. We highlighted gradient update steps to isolate the effect of the additional Trajectory Balance loss term in Equation 18, given that DiffAb and AbFlowNet have almost the same architecture and training data. We find that by simply adding the TB loss term, AbFlowNet improves CDR binding energy by 24.8%. As such, the TB loss term essentially constrains the generation space to CDRs with low binding energy.
> We agree that FLOPs is a better measurement of training compute requirements. Given the review timeline, we report training time for now. Training AbFlowNet for 200K steps required ~45 hours compared to ~27 hours for DiffAb. This is mainly due to the need to sample entire trajectories to determine the TB loss term. We note that online RL methods are likely to be an order of magnitude more expensive due to the need to sample, relax, and score thousands of new CDRs, possibly for multiple iterations, as is common in DPO.

---

> ### Author Response · Authors · 2025-11-21
> **Response to Reviewer Rcth (Part 2 of 2)**
>
> 5. **Novelty of Denoising Trajectory as Path Contribution**
>
> Thank you for suggesting the paper, we will certainly include it in the final manuscript. We agree that denoising trajectories are naturally conceptualized as an RL episode. Our formulation differs in the specifics: AbFlowNet carries out three co-conditioned denoising trajectories in parallel, of which two are continuous (position and orientation generation) and one is discrete (amino acid generation).
> Furthermore, the suggested paper utilizes PPO, which is an online RL algorithm and has the same weaknesses as DPO, namely overfitting to energy estimators and computational cost.
>
>
> 6. **Hyperparameter w in Equation 18**
>
> This is an excellent observation. Intuitively, we want to prevent the model from over-optimizing the binding energy at the expense of diffusion reconstruction. We discuss this in detail in Appendix B.2. An intuitive heuristic is that w should be set such that w · L_TB is comparable in magnitude to L_type, L_pos, and L_ori, to ensure balanced optimization. We transcribe Figure 4 of the manuscript regarding optimizing w below:
>
> | CDR | LR | AAR (↑) | RMSD (↓) |
> | :--- | :--- | :--- | :--- |
> | **H_CDR1** | DiffAb | 64.23 | 1.153 |
> | | 1e-07 | 63.99 | 1.046* |
> | | 5e-07 | **65.81*** | 1.128* |
> | | 1e-06 | 63.49 | **0.974*** |
> | | 5e-06 | 65.05* | 0.993* |
> | | 1e-05 | 65.65* | 1.054* |
> | | 5e-05 | 65.19* | 1.237 |
> | **H_CDR2** | DiffAb | 35.87 | 1.095 |
> | | 1e-07 | 33.45 | 0.942* |
> | | 5e-07 | 35.05 | 0.897* |
> | | 1e-06 | **38.06*** | 0.848* |
> | | 5e-06 | 36.56* | **0.840*** |
> | | 1e-05 | 35.18 | 0.867* |
> | | 5e-05 | 33.42 | 1.137 |
> | **H_CDR3** | DiffAb | 24.34 | 3.236 |
> | | 1e-07 | 24.35* | 3.331 |
> | | 5e-07 | 24.00 | 3.410 |
> | | 1e-06 | 25.08* | 3.194* |
> | | 5e-06 | 25.04* | **3.117** |
> | | 1e-05 | **25.27*** | 3.184* |
> | | 5e-05 | 24.36* | 3.399 |
> | **L_CDR1** | DiffAb | 53.69 | 1.153 |
> | | 1e-07 | 54.72* | 1.156 |
> | | 5e-07 | 55.24* | 1.063* |
> | | 1e-06 | **55.62*** | **0.974*** |
> | | 5e-06 | 55.50* | 1.289 |
> | | 1e-05 | 55.42* | 1.014* |
> | | 5e-05 | 54.45* | 0.974* |
> | **L_CDR2** | DiffAb | 50.46 | 0.795 |
> | | 1e-07 | 50.16 | 0.723* |
> | | 5e-07 | 50.33 | 0.759* |
> | | 1e-06 | **54.09*** | **0.782*** |
> | | 5e-06 | 52.40* | 0.785* |
> | | 1e-05 | 51.75* | 0.759* |
> | | 5e-05 | 50.71* | 0.916 |
> | **L_CDR3** | DiffAb | 44.87 | 3.840 |
> | | 1e-07 | 45.59* | **1.251*** |
> | | 5e-07 | 43.91 | 1.342* |
> | | 1e-06 | 44.68 | 1.310* |
> | | 5e-06 | **45.34*** | 1.266* |
> | | 1e-05 | 45.31* | 1.309* |
> | | 5e-05 | 44.52 | 1.434* |
>
> Results better than DiffAb are marked with an asterisk (*) and the best value is in bold.
>
> We hope these clarifications address your concerns and demonstrate the advantages of AbFlowNet's approach.

---

### Official Review · Reviewer_xeug · 2025-10-27

**Soundness:** 3
**Presentation:** 3
**Contribution:** 2
**Rating:** 6
**Confidence:** 4

**Summary:**

This paper proposed AbFlow, a combination of diffusion models and GFlowNet for antibody design. AbFlow interprets each diffusion step as a GFlowNet state, and use the trajectory balance loss to propagate Rosetta binding energy rewards. This model improves upon AAR, RMSD, and top-1 Rosetta binding energy.

**Strengths:**

- This model unifies data and physical prior through the combination of diffusion and GFlowNet.
- It is more efficient than previous preference optimization-based methods which require collection of massive training pairs.
- It demonstrates performance improvement compared to previous antibody design models (GNN-based and diffusion-based).

**Weaknesses:**

- Evaluation metrics are outdated. Only Rosetta scores are computed which is however noisy and sensitive to minor perturbation. A more reliable evaluation method that has  been used more recently is using AlphaFold to predict the complex structure and compare the RMSD between generation and AlphaFold prediction.

**Questions:**

- Since Rosetta is very noise-sensitive, is it possible to use AlphaFold quality predictor such as PLDDT, which serves as a good proxy for binding in previous studies (e.g. BindCraft, EvoBind), as the reward?

---

> ### Author Response · Authors · 2025-11-21
>
> Thank you for the positive review and your suggestions regarding AlphaFold3-based metrics.
>
> **Clarification on Metrics and the Use of Rosetta**
> As a binding energy optimization framework, we have emphasized binding energy and total energy improvements (Table 1) in the main text. We report per-CDR AAR and RMSD metrics for both test datasets in Appendices A and D, demonstrating that our joint optimization preserves reconstruction quality—unlike the RL-based AbDPO framework.
>
> We agree that Rosetta produces noisy binding energy estimates. We chose Rosetta primarily for consistency with baseline methods (AbDPO, AbNovo, IgGM).
>
> As discussed in our Discussion section (L 446), AbFlowNet does not strictly require an energy estimator and could be trained using wet-lab measurements alone. Unlike online RL methods that need binding energies for newly-generated CDRs, AbFlowNet only requires energies for the training dataset. We used an estimator because wet-lab measurements are not yet available for all complexes in SAbDab.
>
> **RMSD Against AlphaFold-Predicted Structures for Evaluation**:
> We fully appreciate the motivation for using AlphaFold3 as a more stable structural reference. However, AlphaFold3 predicts complete antigen–antibody complexes from sequence alone and currently does not support fixing the antigen and antibody frameworks while predicting only the redesigned CDR loop. As a result, the predicted backbone of the complex differs slightly from the experimental structure, and this drift propagates into the CDR region. Even for the reference 5mes structure, the AlphaFold3-predicted CDR-H3 differs from the experimental loop with an RMSD of 0.252 Å after alignment. Because AbFlowNet conditions on fixed experimental structures, comparing its outputs to AlphaFold3 structures would conflate model quality with AlphaFold3’s own structural deviations.
>
> One alternative would be to replace all test structures with AlphaFold3 predictions and condition AbFlowNet on those instead. IgGM [1] employs a similar approach. However, doing this would effectively integrate AlphaFold3 into the generation pipeline rather than using it as an unbiased evaluation metric.
>
> **AlphaFold pLDDT as a Reward:**
> We agree that AlphaFold quality indicators such as pLDDT are promising as additional reward components. AbFlowNet is reward-agnostic: any scalar or weighted combination of rewards can be incorporated into the trajectory balance objective by replacing R in Eq. 17. Because rewards are precomputed for the fixed training set, integrating pLDDT is computationally feasible and would not require running AlphaFold during inference. Given the success of pLDDT in recent docking studies [2], we see this as a promising direction for future extensions of AbFlowNet.
>
> [1] Wang et al. IgGM: A Generative Model for Functional Antibody and Nanobody Design, ICLR 2025
>
> [2] Joubbi el al. Enhancing antibody-antigen interaction prediction with atomic flexibility. PLoS Comput Biol 21(10): e1013576.

---

### Official Review · Reviewer_uVAT · 2025-10-27

**Soundness:** 3
**Presentation:** 2
**Contribution:** 3
**Rating:** 6
**Confidence:** 4

**Summary:**

The authors reframe the generative design of antibodies by integrating (current) diffusion models with the GFlowNet framework, such that the diffusion step is a state in the GFlowNet framework. This allows them to jointly optimize standard diffusion loss and binding energy in the training procedure, unifying diffusion and reward optimisation in a single procedure, as a competitive approach  to reinforcement learning in which binding energy is based on unreliable estimators. As binding energy can only be computed after complete denoising, they use the trajectory balance objective (TB) to propagate rewards back through the diffusion process. As a consequence, the related rewards for transitions in GFlowNet can be precomputed for each CDR in the training dataset. The resulting method is competitive with a current online reinforcement strategy to post-train a diffusion model (DiffAb) to optimize CDR binding energy.

**Strengths:**

- nice idea to integrate GFlowNet framework with a diffusion model to enforce binding energy constraints in training of generative antibody designs
- clever use of the trajectory balance objective

**Weaknesses:**

- parts are repetitive (intro/related work) wheras other (critical) parts where too condensed (trajectory balance)
- solution relies on trajectory balance objective, which has been proposed in earlier work
- somewhat limited set of metrics used in main text (RMSD,AAR...); comparison on these other metrics is also not trivial/conclusive
- could have expanded the results more top gain more insight in performance, eg not restricting to top-1 metrics but also include distributions of generated designs
- how would you include multiple objectives into this framework?
- Fig 3, explain shading of structure (be more clear on visual differences, instead of only adressing DG)

**Questions:**

Please address points of identified weakness (most textual, but 4th point also with additional figures/tables).

---

> ### Author Response · Authors · 2025-11-21
>
> Thank you for the positive review. We address your concerns below.
>
> **Distributions of Generated Designs:**
> Thank you for this excellent suggestion. We agree that top-1 metrics alone do not fully capture AbFlowNet's capabilities. We have added mean and standard deviation statistics for all CDRs on the RAbD dataset in ***Appendix I of our revised manuscript***, along with distribution plots for several randomly sampled complexes.
>
> **Interpretation of Figure 3 and Additional Visualization:**
> In Figure 3, the globular structure represents the antigen's molecular surface, while the ribbon structure shows the antibody with the CDR-H3 region highlighted. The arrows indicate the fixed antibody sequence boundaries. We aimed to illustrate how well the CDR aligns with the antigen's grooves and cavities which is a determinant of binding energy.
> We acknowledge that Figure 3 is challenging to interpret as a 2D view of a 3D structure. We have therefore added color-coded visualizations with detailed interpretations in ***Appendix J of our revised manuscript*** for improved clarity.
>
> **Including Multiple Objectives:**
> Like other RL methods, GFlowNet can incorporate reward signals from arbitrary sources via weighted summation (replacing R in Eq. 17). This framework naturally accommodates objectives such as evolutionary plausibility [1] assessed by Protein Language Models, or geometry-aware design to minimize side-chain collisions.
>
> **Clarification on Metrics:**
> As a binding energy optimization framework, AbFlowNet emphasizes binding energy and total energy improvements (Table 1). We report per-CDR AAR and RMSD metrics for both test datasets in Appendices A and D, demonstrating that our joint optimization preserves reconstruction quality, unlike the RL-based AbDPO framework which uses the same neural architecture as our work.
>
> **Writing Suggestions:**
> Thank you for identifying specific areas needing clarification. We deeply appreciate it. We will surely address these in future revisions.
>
> [1] Lin et al., 2022. Language models of protein sequences at the scale of evolution enable accurate structure prediction.

---

> > ### Author Response · Authors · 2025-11-29
> > **Energy Distribution of Generated CDRs Part 1**
> >
> > We have copied Table 6 from Appendix I of our revised manuscript for ease of reviewing. Across all complexes, the AbFlowNet-generated CDRs exhibit right-skewed, heavy-tailed energy distributions. The bulk of the generated CDRs have energy values close to the reference structure, while only a small proportion fall into the high-energy tail.
> >
> > | Complex | Energy of Complex E_complex (↓) | | | Binding Energy CDR-Ag ΔG (↓) | | | Total CDR Energy E_total (↓) | | |
> > |---------|---------|---------|---------|---------|---------|---------|---------|---------|---------|
> > | | **Min** | **Mean ± Std** | **Ref** | **Min** | **Mean ± Std** | **Ref** | **Min** | **Mean ± Std** | **Ref** |
> > | 1a14_H_L_N | 465.24 | 739.27 ± 349.05 | 1829.81 | -23.46 | 5.76 ± 154.80 | 185.48 | 18.46 | 151.25 ± 168.06 | 56.52 |
> > | 1a2y_B_A_C | -695.23 | -643.06 ± 86.15 | -412.00 | -20.92 | -15.31 ± 2.99 | 17.32 | -9.88 | 10.63 ± 12.45 | -13.09 |
> > | 1fe8_H_L_A † | 66.58 | 2244.82 ± 1279.23 | -300.06 | 151.96 | 1998.45 ± 1199.38 | 62.53 | 212.53 | 1444.82 ± 731.12 | 29.13 |
> > | 1ic7_H_L_Y † | 94.59 | 920.13 ± 702.52 | 29.49 | 54.26 | 250.78 ± 308.75 | 75.85 | 54.36 | 541.44 ± 397.55 | 17.72 |
> > | 1iqd_B_A_C | -591.29 | -464.97 ± 75.50 | -60.88 | -37.13 | 16.03 ± 35.73 | 206.79 | -1.97 | 33.88 ± 21.23 | 41.34 |
> > | 1n8z_B_A_C | 64.03 | 258.41 ± 166.25 | 2867.82 | 10.21 | 82.56 ± 82.50 | 188.81 | 8.13 | 95.22 ± 72.46 | 34.61 |
> > | 1ncb_H_L_N | 567.36 | 1235.35 ± 781.95 | 1602.17 | -4.30 | 128.75 ± 273.55 | 188.93 | 27.20 | 241.38 ± 337.78 | 31.78 |
> > | 1osp_H_L_O | -396.07 | -69.62 ± 279.88 | -10.07 | -18.15 | 61.66 ± 183.67 | 75.33 | 20.21 | 149.73 ± 135.03 | -2.64 |
> > | 1uj3_B_A_C | -611.02 | -250.98 ± 309.79 | -214.20 | -24.65 | 26.25 ± 58.20 | 54.90 | -1.91 | 36.66 ± 38.08 | -4.91 |
> > | 2adf_H_L_A | -519.56 | -283.61 ± 196.12 | -591.05 | -16.85 | 54.62 ± 127.52 | 25.70 | 10.47 | 66.50 ± 34.71 | -12.68 |
> > | 2b2x_H_L_A | -314.15 | -26.41 ± 231.04 | 375.35 | -17.12 | 24.69 ± 75.21 | 117.86 | 11.17 | 70.64 ± 60.82 | 6.81 |
> > | 2cmr_H_L_A | -581.82 | -355.59 ± 240.43 | 131.70 | -18.15 | 90.94 ± 169.18 | 106.36 | 6.15 | 80.40 ± 77.97 | -3.14 |
> > | 2dd8_H_L_S | -17.64 | 230.40 ± 155.19 | 424.31 | -93.32 | -8.70 ± 20.69 | 181.95 | -1.80 | 38.35 ± 36.71 | 74.37 |
> > | 2ghw_B_b_A | -501.10 | -373.60 ± 151.76 | -124.77 | -33.58 | -11.66 ± 31.11 | 84.61 | -0.74 | 39.36 ± 67.84 | 15.03 |
> > | 2vxt_H_L_I | -371.27 | 686.03 ± 1004.31 | -426.22 | 26.07 | 599.32 ± 838.56 | 32.43 | 25.65 | 641.44 ± 579.75 | -5.41 |
> > | 2xqy_G_L_A † | -805.28 | -673.89 ± 217.51 | -61.00 | -20.04 | -2.50 ± 8.44 | 33.51 | -5.38 | 36.07 ± 32.74 | -2.43 |
> > | 2xwt_A_B_C | -793.86 | -472.31 ± 246.79 | -614.54 | 16.28 | 132.46 ± 60.82 | 21.49 | 2.00 | 66.14 ± 51.55 | -14.67 |
> > | 2ypv_H_L_A | -854.85 | -705.91 ± 116.26 | -589.38 | -18.71 | 10.92 ± 24.19 | 37.34 | 7.40 | 52.64 ± 39.39 | 5.15 |
> > | 3bn9_D_C_B | -406.84 | 1233.60 ± 1772.78 | 256.31 | -16.47 | 815.04 ± 1437.08 | 299.53 | 52.33 | 1029.08 ± 985.85 | 133.68 |
> > | 3cx5_J_K_E | -175.80 | 154.47 ± 320.10 | 265.42 | -14.12 | 5.47 ± 25.08 | 60.79 | 21.83 | 171.98 ± 154.79 | -10.61 |
> > | 3h3b_c_C_B | -458.08 | -233.14 ± 277.84 | -11.97 | 2.34 | 6.65 ± 4.86 | 41.78 | 11.24 | 96.08 ± 84.23 | 10.90 |
> > | 3hi6_X_Y_B | -580.80 | -180.53 ± 502.57 | -374.75 | -17.26 | 138.10 ± 403.76 | 49.91 | 20.04 | 181.51 ± 198.83 | 5.80 |
> > | 3k2u_H_L_A | 2288.98 | 2402.07 ± 83.48 | 2858.45 | -26.46 | 3.46 ± 19.54 | 110.99 | 9.60 | 79.59 ± 58.88 | 18.79 |
> > | 3l95_B_A_X | -92.20 | 164.80 ± 119.44 | 219.03 | -12.62 | 27.84 ± 49.06 | 149.92 | 12.96 | 85.10 ± 45.78 | 2.93 |
> > | 3mxw_H_L_A | -483.43 | -339.37 ± 120.16 | -485.02 | -17.19 | -0.88 ± 16.07 | 26.31 | 6.20 | 55.40 ± 36.81 | -1.16 |
> > | 3nid_H_L_AD | -1627.05 | -1420.20 ± 157.44 | -1232.12 | -22.66 | 30.28 ± 25.74 | -1.90 | 16.14 | 119.02 ± 81.13 | -14.81 |
> > | 3o2d_H_L_A | -647.48 | -320.46 ± 339.92 | 99.11 | -108.43 | 19.95 ± 87.34 | 116.32 | 15.60 | 156.66 ± 162.21 | 0.16 |
> >
> > **Note:** † denotes complexes that could not be processed with PyRosetta FastRelax.

---

> > > ### Author Response · Authors · 2025-11-29
> > > **Energy Distribution of Generated CDRs Part 2**
> > >
> > > | Complex | Energy of Complex E_complex (↓) | | | Binding Energy CDR-Ag ΔG (↓) | | | Total CDR Energy E_total (↓) | | |
> > > |---------|---------|---------|---------|---------|---------|---------|---------|---------|---------|
> > > | | **Min** | **Mean ± Std** | **Ref** | **Min** | **Mean ± Std** | **Ref** | **Min** | **Mean ± Std** | **Ref** |
> > > | 3rkd_H_L_A | -559.58 | 546.00 ± 1289.63 | -293.93 | -26.78 | 237.52 ± 414.37 | -18.92 | 27.52 | 541.60 ± 615.68 | -2.53 |
> > > | 3s35_H_L_X | -645.40 | -563.05 ± 80.61 | -123.47 | -20.89 | 14.90 ± 62.73 | 65.90 | -1.80 | 31.35 ± 29.16 | 1.75 |
> > > | 3uzq_A_a_B | -544.58 | 442.73 ± 2213.79 | 27501.00 | -42.78 | -2.51 ± 123.08 | - | 7.26 | 191.34 ± 260.35 | -15.86 |
> > > | 3w9e_A_B_C | -545.74 | -72.71 ± 407.60 | -129.53 | 2.03 | 100.45 ± 240.09 | 88.40 | 36.48 | 210.11 ± 162.17 | -2.35 |
> > > | 4cmh_B_C_A | -836.91 | -441.40 ± 309.48 | -591.38 | -30.57 | 16.98 ± 47.83 | 51.93 | 18.21 | 187.31 ± 134.80 | -10.43 |
> > > | 4dtg_H_L_K | -300.15 | -99.86 ± 192.58 | -125.16 | -6.02 | 18.94 ± 25.45 | 58.17 | 9.78 | 127.47 ± 144.68 | 15.80 |
> > > | 4dvr_H_L_G | -366.66 | -253.46 ± 123.38 | 146.55 | -19.77 | -2.01 ± 38.03 | 58.58 | -8.60 | 40.92 ± 53.03 | -1.64 |
> > > | 4etq_H_L_C | -564.55 | -337.49 ± 177.81 | -161.47 | -0.33 | 115.77 ± 50.70 | 73.30 | -3.46 | 41.51 ± 38.69 | -2.37 |
> > > | 4ffv_D_C_B | -610.13 | -399.67 ± 131.91 | 347.73 | -20.34 | -6.64 ± 13.72 | 90.90 | 6.48 | 31.31 ± 28.74 | 26.73 |
> > > | 4fqj_H_L_A | -191.74 | 1270.81 ± 1615.52 | 329.45 | 2.09 | 662.42 ± 998.74 | 130.07 | 75.53 | 732.24 ± 771.87 | 21.99 |
> > > | 4g6j_H_L_A | -284.25 | -177.17 ± 106.34 | -278.38 | 2.98 | 39.60 ± 32.32 | 97.63 | -4.55 | 43.90 ± 31.31 | 7.59 |
> > > | 4g6m_H_L_A | -727.04 | -621.49 ± 122.25 | -416.44 | -16.21 | 3.93 ± 29.46 | 36.74 | -7.36 | 42.04 ± 41.27 | -7.57 |
> > > | 4h8w_H_L_G | -765.55 | -525.18 ± 167.65 | -500.04 | -22.41 | 62.50 ± 97.84 | 49.32 | 11.28 | 104.24 ± 83.04 | 1.33 |
> > > | 4ki5_E_F_M | -193.51 | 289.45 ± 524.73 | -175.88 | -22.73 | 129.03 ± 232.47 | 45.83 | 22.34 | 211.71 ± 275.09 | -6.51 |
> > > | 4lvn_C_B_A | -650.42 | -148.01 ± 571.38 | -212.13 | -25.83 | 94.06 ± 208.54 | 68.50 | 20.18 | 170.38 ± 233.80 | 29.01 |
> > > | 4ot1_H_L_A | -377.87 | 1924.70 ± 2775.83 | -332.76 | -5.03 | 489.36 ± 1109.65 | 62.94 | 95.39 | 1792.68 ± 2029.26 | -2.69 |
> > > | 4qci_B_A_D | -299.23 | -130.16 ± 123.36 | -76.90 | -17.54 | 53.22 ± 52.44 | 37.63 | 5.22 | 63.86 ± 63.54 | 14.68 |
> > > | 4xnq_B_A_D | 190.39 | 582.77 ± 311.38 | 238.64 | -10.39 | 54.82 ± 71.60 | 34.20 | 41.31 | 271.33 ± 197.50 | -9.83 |
> > > | 4ydk_H_L_G | -754.23 | 1338.02 ± 2353.61 | -617.74 | -25.80 | 841.97 ± 1496.09 | 22.24 | 73.71 | 1361.75 ± 1483.04 | -23.96 |
> > > | 5b8c_B_A_C | -405.83 | -89.10 ± 259.21 | -322.93 | 15.95 | 119.52 ± 80.35 | 90.55 | 20.50 | 120.70 ± 107.06 | 0.10 |
> > > | 5bv7_C_B_A | 196.59 | 920.92 ± 828.95 | 1230.31 | 21.23 | 172.09 ± 217.10 | 113.39 | 49.49 | 447.16 ± 464.02 | 44.74 |
> > > | 5d93_C_B_A † | -479.00 | 805.38 ± 1122.77 | -541.29 | 25.53 | 729.50 ± 854.56 | 22.08 | 29.29 | 860.37 ± 716.65 | -0.84 |
> > > | 5d96_J_I_D | -197.67 | -50.32 ± 191.34 | 295.45 | -15.31 | 23.77 ± 103.76 | 87.44 | 1.94 | 91.77 ± 100.97 | 11.20 |
> > > | 5en2_A_B_C | -681.11 | -257.33 ± 683.81 | -532.16 | -27.48 | 44.84 ± 220.13 | 14.59 | 20.79 | 280.10 ± 378.01 | -12.95 |
> > > | 5f9o_H_L_G | -748.61 | -416.83 ± 315.42 | 67.19 | -39.86 | 13.56 ± 64.36 | 88.09 | 12.70 | 168.53 ± 152.44 | 4.09 |
> > > | 5ggs_A_B_Z | -412.43 | -123.40 ± 249.30 | -137.59 | -12.58 | 52.22 ± 82.59 | 56.20 | 11.03 | 129.07 ± 135.37 | -8.39 |
> > > | 5hi4_H_L_A | -508.21 | -398.51 ± 114.40 | -215.09 | -20.36 | 8.50 ± 96.39 | -24.37 | 6.73 | 55.59 ± 33.81 | 8.09 |
> > > | 5j13_C_B_A | -475.93 | -298.67 ± 151.07 | -365.81 | -18.03 | 1.64 ± 19.68 | 16.54 | 16.60 | 116.51 ± 103.25 | 5.91 |
> > > | 5l6y_H_L_C | -487.69 | -190.69 ± 319.95 | -360.32 | -29.21 | 20.40 ± 68.66 | -3.57 | 21.19 | 164.42 ± 174.97 | 6.56 |
> > > | 5mes_H_L_A | -594.11 | -422.81 ± 94.28 | -156.46 | -7.90 | -0.74 ± 7.36 | -7.93 | -2.23 | 43.10 ± 35.82 | -9.65 |
> > > | 5nuz_A_B_C | -758.42 | -556.64 ± 239.14 | -674.95 | -31.56 | -1.03 ± 36.37 | -11.37 | 11.88 | 87.82 ± 70.17 | -9.43 |
> > >
> > > **Note:** † denotes complexes that could not be processed with PyRosetta FastRelax.

---

### Official Review · Reviewer_wrxN · 2025-10-29

**Soundness:** 3
**Presentation:** 3
**Contribution:** 2
**Rating:** 4
**Confidence:** 4

**Summary:**

This paper introduces AbFlowNet, a generative framework for structure-based antibody redesign with additional emphasis and consideration of the force-field estimate binding energy.

**Strengths:**

- Authors introduce a method allowing to jointly optimise for standard generative / diffusion objectives and binding energy estimated through force field methods.
- Benchmarking on standard datasets / splits reveals slight improvements across investigated standard metrics (amino acid recovery, RMSD to wild-type conformation).

**Weaknesses:**

- I find the benchmarking to be not convincing, which is reflected in my score. Qualitative example shown in Fig 3 is not clear at all, even for a person with trained structural eye and energy differences are miniscule.

**Questions:**

- Following on the highlighted weakness point - I’d suggest authors to consider showing some properties of generated sequences and how they overlap with some reference distribution. There are multiple datasets with several binders and non-binders around wild-type sequence with corresponding structure information. Furthermore, highlighting generative abilities of the model on this data (e.g. are affinity improving mutations suggested by the model more frequently compared to other baselines? or, if possible within the framework, is the estimated likelihood of improving variant compared to wild-type better?) would allow to assess the improvements postulated by the authors without direct experimental validation.

---

> ### Author Response · Authors · 2025-11-21
>
> Thank you for the constructive feedback and for engaging closely with our work. We appreciate your assessment of the strengths and weaknesses and have made several updates that address your suggestions and clarify our contributions. Our goal with AbFlowNet is to unify diffusion-based CDR generation with explicit binding energy optimization, providing a framework that integrates structure, generative modeling, and energy guidance in a single training process. Below, we address each of your comments:
>
> **Benchmarking Results**: As a binding-energy–focused optimization framework, our primary emphasis in the main text (Table 1) is on improvements in binding and total energy. AbFlowNet achieves substantial gains over the DiffAb baseline, improving CDR total energy by 24.8% and binding energy by 38.1%. The observed improvements in AAR and RMSD are benefits that arise naturally from unifying diffusion-based reconstruction and GFlowNet-based energy optimization within a single, coherent optimization process. As discussed in Section 6.2, prior work has shown that performing diffusion training followed by a separate energy-optimization stage actually degrades reference-based performance, reducing AAR by 9.96% and increasing RMSD by 0.14 Å.
>
>
> **Additional Qualitative Examples:** Thank you for pointing out that the original qualitative example in Fig. 3 was difficult to interpret, even for a trained structural eye. We appreciate this comment, and in response, we added an ***expanded set of qualitative examples in Appendix J of our revised manuscript*** with clearer structural interpretation. These updated visualizations include hydrophilic and hydrophobic color-coding, surface-contact views, and residue-level packing analyses. They make it much easier to see how AbFlowNet positions CDR-H3 loops with improved interface alignment and more consistent hydrophobic–hydrophilic complementarity, which RMSD and raw ΔG alone do not fully capture. The examples in Appendix J also visualize antigen–antibody contact surfaces and the interactions stabilizing the complex, showing that AbFlowNet achieves improved surface alignment and residue-level interactions compared to DiffAb, with performance comparable to the reference CDRs. We are grateful for your suggestion, which strengthened the clarity and interpretability of our benchmarking.
>
> **Properties of Generated Sequences:** Thank you for the suggestion to examine broader sequence properties. We have added an evolutionary plausibility analysis in ***Appendix K of our revised manuscript*** using the ESM-2 protein language model to assess the naturalness of our generated designs. As shown in the new Figure 10, embeddings of our generated CDRs cluster tightly with reference sequences. This indicates that while we optimize for binding energy, we do not drift into 'adversarial' regions of sequence space; the generated antibodies remain evolutionarily plausible and retain the statistical properties of natural immunoglobulins. This complements our existing AAR and RMSD metrics (Appendices A and D), which are also computed against the reference.
>
> **Mutation Prediction and Likelihood Estimation:** These are excellent suggestions for model assessment; however, diffusion models such as AbFlowNet do not natively support mutation or likelihood estimation without substantial modification. Diffusion models such as DiffAb, AbDPO, and AbflowNet currently generate complete de novo CDRs by starting from a known Gaussian or Uniform distribution and iteratively denoising both the structure and the amino acid sequence. We hypothesize that controlled mutation generation is feasible through partial noising and denoising of wild-type structures. However, our pilot experiments suggest that partially noising a portion of the CDR while keeping the other region fixed is highly out-of-distribution for all the CDR diffusion models. To mitigate this, finetuning is likely required. Similarly, diffusion models directly sample from a learned target distribution rather than estimate the likelihood of a sample being from a target distribution. Recent work [1] has shown that diffusion model likelihoods can be improved through finetuning, suggesting a promising direction for future CDR design applications.
>
> We believe this represents an important direction for the field, as mutation-based evaluation would complement de novo generation capabilities and provide clinically relevant benchmarks.
>
> [1] Zheng et al. ICML 2023. Improved Techniques for Maximum Likelihood Estimation for Diffusion ODEs

---

### Meta-Review · Area_Chair_he73 · 2026-01-06

**Summary:**

This paper proposes AbFlow, which combines diffusion models with GFlowNet for antibody design. The method interprets each diffusion step as a GFlowNet state and applies trajectory balance loss to propagate Rosetta binding energy as a reward signal. The authors report improvements in AAR, RMSD, and top-1 Rosetta binding energy.

**Reviewer Concerns:**

However, several concerns remain: (1) the evaluation metrics are outdated, (2) comparisons with recent SOTA methods are insufficient, and (3) the overall benchmarking is not convincing. While the authors attempted to address these issues in the rebuttal, the core dependence on Rosetta scoring functions is still not adequately resolved, and the main results in the paper remain limited and unpersuasive.

Overall, the evaluation is narrow and relies on metrics whose reliability is questionable, making it difficult to assess whether the method truly surpasses recent approaches. Although the algorithmic idea shows some novelty, I am inclined to recommend rejection at this stage, and would encourage the authors to provide more comprehensive experimental validation (e.g., additional scoring with AlphaFold-based metrics) in future revisions.

**Reviewer Scores:**

I do not expect any substantial changes to the reviewer scores.

---

### Decision · Program_Chairs · 2026-01-26

Reject